# Online Control-Informed Learning

**Zihao Liang**        *liang331@purdue.edu*
*School of Aeronautics and Astronautics*
*Purdue University*

**Tianyu Zhou**        *zhou1043@purdue.edu*
*School of Aeronautics and Astronautics*
*Purdue University*

**Zehui Lu**        *lu846@purdue.edu*
*School of Aeronautics and Astronautics*
*Purdue University*

**Shaoshuai Mou**        *mous@purdue.edu*
*School of Aeronautics and Astronautics*
*Purdue University*

**Reviewed on OpenReview:** *https://openreview.net/forum?id=LDzvZEVl5H*

## Abstract

This paper proposes an Online Control-Informed Learning (OCIL) framework, which employs the well-established optimal control and state estimation techniques in the field of control to solve a broad class of learning tasks in an online fashion. This novel integration effectively handles practical issues in machine learning such as noisy measurement data, online learning, and data efficiency. By considering any robot as a tunable optimal control system, we propose an online parameter estimator based on extended Kalman filter (EKF) to incrementally tune the system in an online fashion, enabling it to complete designated learning or control tasks. The proposed method also improves the robustness in learning by effectively managing noise in the data. Theoretical analysis is provided to demonstrate the convergence of OCIL. Three learning modes of OCIL, i.e. Online Imitation Learning, Online System Identification, and Policy Tuning On-the-fly, are investigated via experiments, which validate their effectiveness.

## 1 Introduction

Informed Machine Learning (IML) (Von Rueden et al., 2021) represents an emerging approach that integrates prior knowledge into the machine learning (ML) process. While classic classification tasks in unsupervised, semi-supervised, or supervised ML primarily focus on extracting patterns from labeled or unlabeled data (LeCun et al., 2015), IML leverages prior knowledge such as physical laws, expert knowledge, or existing models to uncover underlying connections within data (Karniadakis et al., 2021). This integration enables models to produce more reliable and interpretable predictions, especially when dealing with noisy data. This approach is especially advantageous in the domains where theoretical understanding is well-established and thus can guide ML. One notable example of IML is physics-informed machine learning (Wu et al., 2018; Karniadakis et al., 2021; Kashinath et al., 2021), which is particularly valuable for solving partial differential equations for computational fluid dynamics.

Control-informed learning (CIL) is a subset of IML tailored for system control, autonomy, and robotics. This approach merges standard control theory with ML techniques to enhance the capabilities of autonomous systems. The integration leverages the complementary strengths of control and learning. Control theory provides model structures and optimization guidance that enable efficient and reliable algorithms for handling complex tasks. Meanwhile, ML improves these models by learning from data, a capacity that some conventional

control methods lack (Jin et al., 2020; 2021b). This paper aims to tackle learning tasks in autonomous systems that are governed by optimal control (OC) systems. An optimal control system usually consists of dynamics, a control policy, and an objective function. From a unified perspective, learning these components can be understood as addressing a common problem with unknown parameters in different parts of the system and using different loss functions. For example, in learning dynamics, the task involves parameterizing a differential equation, with the loss function measuring the prediction error between the model's output and the target data. In learning policies, the unknown parameters are within the feedback policy and the control objective itself serves as a loss function. When learning control objective functions, the objective is parameterized, and the loss measures the discrepancy between the reproduced trajectory and observed demonstrations.

To tackle these problems, many works in the field of so-called Learning for Dynamics and Control aim to leverage the integration of learning and control but often treat them as separate or sequential tasks. For example, control theories are used for algorithm development and convergence analysis of online unconstrained or constrained optimizations (Casti et al., 2023; Bastianello et al., 2024; Lu et al., 2024); model-based reinforcement learning (Heess et al., 2015; Gu et al., 2016), improves sample efficiency by using dynamics models; Koopman-operator control (Proctor et al., 2018; Abraham & Murphey, 2019; Hao et al., 2024), employs learning to transform nonlinear systems into linear observable space, simplifying control design. In contrast, CIL integrates these processes, allowing learning algorithms to incorporate control insights directly. The integration enables ML and control techniques to perform simultaneously, reducing computational complexity, and improving practical applicability. CIL differentiates itself by utilizing Pontryagin's maximum principle, a foundational concept in OC theory. This principle defines the optimality conditions for the state and input trajectories of an OC system. CIL employs these conditions to provide gradients for machine learning (Jin et al., 2020; 2021b; Böttcher et al., 2022). CIL integrates these gradients directly into its learning process, ensuring that machine learning outcomes are efficient while remaining consistent with established control theories and physical models. This approach enhances both the reliability and accuracy of the results.

## 1.1 Related Work

This section presents existing research on learning various components of an autonomous system and explores related learning frameworks that address these problems from a unified perspective.

**Learning dynamics.** To learn a nonlinear system with possibly noisy measurement, Markov decision-process-based methods are widely used, such as linear regression (Haruno et al., 2001), observation-transition modeling (Finn et al., 2016), latent space modeling (Watter et al., 2015), (deep) neural networks (NN) (Li et al., 2018; Li & Hao, 2018; Han et al., 2019; Zhang et al., 2019; Benning et al., 2019; Liu & Markowich, 2020; Beintema et al., 2023; Pillonetto et al., 2025), Gaussian processes (Deisenroth & Rasmussen, 2011), and transition graphs (Zhang et al., 2018). Despite their widespread use, these methods often must balance data efficiency with prediction accuracy. To improve both metrics, physics-informed learning approaches Lutter et al. (2019); Xu et al. (2020); Saemundsson et al. (2020); Sharma et al. (2023) incorporate physical laws into learning models. Koopman operator theory offers a method for lifting states to an infinite-dimensional linear observable space (Mauroy et al., 2020; Liang et al., 2023; Hao et al., 2023; Liu et al., 2024).

**Learning objective functions.** Objective learning is typically referred to as inverse reinforcement learning (IRL) in the ML community and inverse optimal control (IOC) in the system control community. These methods aim to deduce a control objective function with observed optimal demonstrations. (Brown et al., 2019) The objective function is generally represented as a weighted sum of features (Abbeel & Ng, 2004; Ratliff et al., 2006; Ziebart et al., 2008; Arora & Doshi, 2021). Approaches to find these unknown weights include feature matching (Abbeel & Ng, 2004), maximum entropy (Ziebart et al., 2008), maximum margin (Ratliff et al., 2006), and approximate variational reward imitation learning (Chan & van der Schaar, 2021). As for learning nonlinear parameter mapping of objective functions, prior and system-dependent knowledge is required to further extend the methods above. On the other hand, with system dynamics, IOC aims for efficient learning approaches (Mombaur et al., 2010). For example, some methods (Keshavarz et al., 2011; Jin et al., 2019; 2021a; Jin & Mou, 2021; Liang et al., 2022; 2023) directly calculate unknown weights by minimizing the violation of optimality conditions by the observed demonstration data, which avoids repeatedly solving OC problems.

**Learning control policies.** Learning policies are generally termed reinforcement learning (RL) and OC in the ML and control communities, respectively. In the RL community, there are mainly two streams of research, namely model-free and model-based RL. Model-free RL learns policies by directly interacting with the environment, without using a model of it (Mnih et al., 2013; 2015; Oh et al., 2016). To improve data complexity, model-based RL learns a dynamics model before policy learning (Schneider, 1997; Abbeel et al., 2006; Deisenroth & Rasmussen, 2011; Levine & Abbeel, 2014; Gu et al., 2016). For OC, the first strategy is based on dynamical programming, such as the linear quadratic regulator (LQR) (Scokaert & Rawlings, 1998), which solves the OC problem with linear dynamics and quadratic cost, the linear quadratic Gaussian (Athans, 1971), which combines LQR with a Kalman filter to solve OC problem with linear system affected by Gaussian noise, the iterative linear quadratic regulator (iLQR) (Li & Todorov, 2004), which linearizes the dynamics and quadratizes the value function, and differential dynamical programming, which quadratizes the dynamics and value function. Another strategy relies on Pontryagin's maximum/minimal principle (PMP) (Pontryagin, 2018), such as shooting methods (Bock & Plitt, 1984) and collocation methods (Patterson & Rao, 2014). These open-loop methods are further improved by closed-loop methods such as model predictive control (MPC) (Schwenzer et al., 2021), which repeatedly solves an OC problem over a finite horizon to generate control inputs. Recently, Jin et al. (2020) proposed a framework for learning an optimal policy based on differentiating Pontryagin's Maximum Principle.

Many research studies also focus on incremental policy tuning. One of the most popular tracks is transfer learning, which exploits the generalization of existing knowledge so that it can be transferred across different domains (Taylor & Stone, 2009). Recently, transfer learning has been implemented to speed up the learning process in RL (Taylor & Stone, 2009). Another popular method is behavior cloning (Torabi et al., 2018; Czarnecki et al., 2019; Sasaki & Yamashina, 2021). In the control community, tuning OC systems initially refers to neighboring extremal optimal control (NEOC) (Bryson, 1975; Ghaemi et al., 2009). There are other popular methods including adaptive control (Ioannou & Sun, 2012; Bertsekas, 2022; Luo et al., 2023; Guo & Pan, 2023), which adjusts its parameters in real-time to maintain optimal performance, even in the presence of uncertainties or changes in system dynamics, and Bayesian optimization for controller tuning, (Khosravi et al., 2021; Sorourifar et al., 2021; Berkenkamp et al., 2023).

To sum up, there are numerous existing methods focused on individual tasks. These approaches are effective when only one component of the system is unknown. However, in many real-world scenarios, multiple components may be unavailable or uncertain simultaneously. For instance, in autonomous driving, the dynamics of the vehicle may be unknown due to changes in road conditions or vehicle wear and tear. Simultaneously, the control policy may also be unavailable due to a lack of predefined rules or data. In such cases, existing methods often fall short, as they are not designed to handle the joint learning of multiple interdependent components, limiting their applicability in more complex or incomplete systems.

**Unified learning frameworks.** Several studies have explored unified learning frameworks to tackle learning challenges in autonomous systems. These approaches integrate an implicit planner directly within the policy (Okada et al., 2017; Pereira et al., 2018; Amos et al., 2018; Srinivas et al., 2018). The main challenge in these methods is learning the OC system, which is very similar to the goal of this work. (Okada et al., 2017; Pereira et al., 2018) learn a path-integral OC system (Kappen, 2005), which is a special class of OC systems. (Srinivas et al., 2018) learns an OC system in a latent space. These methods rely on an "unrolling" strategy to make differentiation easier. Essentially, they treat solving an OC problem as an "unrolled" computational graph created by applying gradient descent repeatedly. This allows automatic differentiation tools (Abadi et al., 2016) to be used directly. This approach faces a few challenges: (i) it requires storing all intermediate steps, making it memory-intensive, and (ii) the accuracy of the gradients depends on how many steps are included in the graph, leading to a trade-off between computational cost and accuracy. To tackle these issues, Amos et al. (2018) proposed a differentiable MPC framework. In the forward pass, it uses an LQR approximation of the OC system, and in the backward pass, gradients are computed by differentiating this LQR approximation. This framework has a major challenge: differentiating LQR requires solving a large linear equation, involving the inversion of a matrix with size proportional to the time horizon, making it very costly for long-horizon systems. To address the challenges of the framework mentioned above, Jin et al. (2020) proposed Pontryagin's differential programming (PDP). PDP avoids unrolled computational graphs by only storing the resulting trajectory without concern about how it is solved. Instead of relying on intermediate

LQR approximations, it directly differentiates through Pontryagin's Maximum Principle (PMP) to obtain exact gradients. Furthermore, its backward pass uses an auxiliary control system to obtain the gradient, reducing memory and computational complexity. First, it lacks the ability for online learning, as it relies on gradient descent to update unknown parameters in the OC system, requiring significant computation time to reach convergence. This drawback is particularly problematic in applications like autonomous driving, where quick adaptation to new scenarios is essential for safety and performance. Second, PDP does not account for noisy measurement data, limiting its effectiveness in real-world situations where sensor data is often unreliable or noisy.

## 1.2 Contributions

This paper introduces an online learning framework called Online Control-Informed Learning (OCIL). This framework is designed to be data efficient for various learning and control tasks while providing robustness against noisy data. In this paper, we consider an autonomous system as an OC system, which is parameterized by tunable parameters within different components of the system, including dynamics, policy, and objective function. By tuning the OC system in an online fashion, the proposed OCIL tackles three learning tasks in robotics, namely Online Imitation Learning, Online System Identification, and Policy Tuning On-the-fly. The proposed OCIL consists of two main components, both of which are inspired by control theory. Specifically, the framework first proposes an online parameter estimator based on the classic online state estimation techniques in control theory. The estimator continually updates the parameter estimates in an online fashion as new data becomes available, aiming to minimize a cumulative loss defined for a specific task. To do so, the gradient information for the loss with respect to the tunable parameter is required. Therefore, OCIL employs a gradient generator (GG) based on Pontryagin Differential Programming in OC theory to calculate the exact gradient.

**Notations.** $\|\cdot\|$ denotes the Euclidean norm. Given a matrix $A \in \mathbb{R}^{n \times m}$, let $A'$ denotes its transpose. For positive integers $n$ and $m$, let $\mathbf{I}_n$ be the $n \times n$ identity matrix; $\mathbf{0}_n \in \mathbb{R}^n$ denotes a vector with all value 0; $\mathbf{0}_{n \times m}$ denotes a $n \times m$ matrix with all value 0. Let $\mathrm{col}\{\boldsymbol{v}_1, \ldots, \boldsymbol{v}_a\}$ denote a column stack of elements $\boldsymbol{v}_1, \ldots, \boldsymbol{v}_a$, which may be scalars, vectors or matrices, i.e. $\mathrm{col}\{\boldsymbol{v}_1, \ldots, \boldsymbol{v}_a\} \triangleq [\boldsymbol{v}_1' \ldots \boldsymbol{v}_a']$.

## 2 Problem Formulation

Consider the following class of OC systems $\Sigma(\boldsymbol{\theta}^*)$, where $\boldsymbol{\theta}^* \in \mathbb{R}^p$ denotes the unknown and constant parameter. The behavior of $\Sigma(\boldsymbol{\theta}^*)$ is determined by minimizing a control objective function:

$$\{\boldsymbol{x}_{1:T}(\boldsymbol{\theta}^*), \boldsymbol{u}_{0:T-1}(\boldsymbol{\theta}^*)\} = \underset{\boldsymbol{x}_{1:T}, \boldsymbol{u}_{0:T-1}}{\arg\min} \quad J(\boldsymbol{x}_{0:T}, \boldsymbol{u}_{0:T-1}, \boldsymbol{\theta}^*) = \sum_{t=0}^{T-1} c(\boldsymbol{x}_t, \boldsymbol{u}_t, \boldsymbol{\theta}^*) + h(\boldsymbol{x}_T, \boldsymbol{\theta}^*) \quad \text{(1a)}$$

$$\text{s.t.} \quad \boldsymbol{x}_{t+1} = \boldsymbol{f}(\boldsymbol{x}_t, \boldsymbol{u}_t, \boldsymbol{\theta}^*), \quad \text{with } \boldsymbol{x}_0 \text{ given.} \quad \text{(1b)}$$

where $t = 0, 1, 2, \cdots, T$ is the time index with $T$ being the final time; $\boldsymbol{x}_t \in \mathbb{R}^n$ and $\boldsymbol{u}_t \in \mathbb{R}^m$ denote the system state and control input, respectively; $\boldsymbol{x}_{0:T}(\boldsymbol{\theta}^*) \triangleq \mathrm{col}\{\boldsymbol{x}_0(\boldsymbol{\theta}^*), \cdots, \boldsymbol{x}_T(\boldsymbol{\theta}^*)\}$ and $\boldsymbol{u}_{0:T-1}(\boldsymbol{\theta}^*) \triangleq \mathrm{col}\{\boldsymbol{u}_0(\boldsymbol{\theta}^*), \cdots, \boldsymbol{u}_{T-1}(\boldsymbol{\theta}^*)\}$ denote the states and inputs trajectory given parameter $\boldsymbol{\theta}^*$, respectively; $\boldsymbol{x}_t^*(\boldsymbol{\theta}^*)$ and $\boldsymbol{u}_t^*(\boldsymbol{\theta}^*)$ denote the state and input given $\boldsymbol{\theta}^*$ at time $t$ respectively; $\boldsymbol{f} : \mathbb{R}^n \times \mathbb{R}^m \times \mathbb{R}^p \to \mathbb{R}^n$ denotes a twice-differentiable time-invariant system dynamics; $c : \mathbb{R}^n \times \mathbb{R}^m \times \mathbb{R}^p \mapsto \mathbb{R}$ and $h : \mathbb{R}^n \times \mathbb{R}^p \mapsto \mathbb{R}$ denote running cost the final cost, respectively, both of which are assumed to be twice-differentiable.

**Remark 1.** *Including the parameter $\theta^*$ in the system dynamics allows for the representation of both partially known and completely unknown dynamics. For partially known dynamics, it is parameterized via a known physical dynamic model with unknown physical parameters. For example, this could be a quadrotor dynamics with known structure and unknown inertia and mass (Wang et al., 2014; Jin et al., 2020; Revach et al., 2022). In the case of completely unknown dynamics, parameterization is done by neural networks. In this case, the neural network captures the evolution of the state, where the parameter $\theta^*$ represents the weights and biases of the neural network (Kumpati et al., 1990; Lewis et al., 1998; Nelles & Nelles, 2020).*

For notation simplicity, we define the unknown trajectory of the optimal control system $\Sigma(\boldsymbol{\theta}^*)$ as

$$\boldsymbol{\xi}(\boldsymbol{\theta}^*) \triangleq \mathrm{col}\{\boldsymbol{x}_{0:T}(\boldsymbol{\theta}^*), \boldsymbol{u}_{0:T-1}(\boldsymbol{\theta}^*)\} \in \mathbb{R}^{(T+1)n+Tm} \quad \text{(2)}$$

The goal of this paper is to estimate $\boldsymbol{\theta}^*$. Define $\hat{\boldsymbol{\theta}} \in \mathbb{R}^p$ as an arbitrary estimation of $\boldsymbol{\theta}^*$. Then for estimation purposes, a copy, $\Sigma(\hat{\boldsymbol{\theta}})$, of the autonomous system $\Sigma(\boldsymbol{\theta}^*)$ can be proposed by replacing $\boldsymbol{\theta}^*$ with $\hat{\boldsymbol{\theta}}$ in (1), i.e.,

$$\{\boldsymbol{x}_{1:T}(\hat{\boldsymbol{\theta}}), \boldsymbol{u}_{0:T-1}(\hat{\boldsymbol{\theta}})\} = \underset{\boldsymbol{x}_{1:T}, \boldsymbol{u}_{0:T-1}}{\arg\min} \quad J(\boldsymbol{x}_{0:T}, \boldsymbol{u}_{0:T-1}, \hat{\boldsymbol{\theta}}) = \sum_{t=0}^{T-1} c(\boldsymbol{x}_t, \boldsymbol{u}_t, \hat{\boldsymbol{\theta}}) + h(\boldsymbol{x}_T, \hat{\boldsymbol{\theta}}) \quad (3a)$$

$$\text{s.t.} \qquad \boldsymbol{x}_{t+1} = \boldsymbol{f}(\boldsymbol{x}_t, \boldsymbol{u}_t, \hat{\boldsymbol{\theta}}), \text{ with } \boldsymbol{x}_0 \text{ given.} \qquad (3b)$$

At each time $t$, a noisy measurement $\boldsymbol{O}_t \in \mathbb{R}^r$ is observed, where

$$\boldsymbol{O}_t = \boldsymbol{h}(\boldsymbol{\xi}_t(\boldsymbol{\theta}^*)) + \boldsymbol{v}_t. \qquad (4)$$

Here, $\boldsymbol{h} : \mathbb{R}^{n+m} \mapsto \mathbb{R}^r$ denotes a twice-differentiable measurement function; $\boldsymbol{\xi}_t(\boldsymbol{\theta}^*) = \{\boldsymbol{x}_t^*(\boldsymbol{\theta}^*), \boldsymbol{u}_t^*(\boldsymbol{\theta}^*)\}$; $\boldsymbol{v}_t \sim \mathcal{N}(\boldsymbol{0}_r, \boldsymbol{R}_t)$ denotes the measurement noise which is a multivariate Gaussian, with $\boldsymbol{R}_t \in \mathbb{R}^{r \times r}$ being the covariance matrices of the measurement noise.

With the measurement equation (4) defined, this paper considers a signed residual function:

$$\boldsymbol{l}(\boldsymbol{\xi}_t(\hat{\boldsymbol{\theta}}), \boldsymbol{O}_t) = \boldsymbol{O}_t - \boldsymbol{h}(\boldsymbol{\xi}_t(\hat{\boldsymbol{\theta}})) \in \mathbb{R}^r. \qquad (5)$$

Then, the performance of the entire trajectory can be evaluated by a cumulative loss which is assumed to be twice-differentiable:

$$L(\boldsymbol{\xi}(\hat{\boldsymbol{\theta}})) = \sum_{t=0}^{T} \|\boldsymbol{l}(\boldsymbol{\xi}_t(\hat{\boldsymbol{\theta}}), \boldsymbol{O}_t)\|^2. \qquad (6)$$

The *problem of interest* is to develop an online method to update the estimation $\hat{\boldsymbol{\theta}}_t \in \mathbb{R}^p$ of $\boldsymbol{\theta}^*$ at every time $t$, such that its trajectory $\boldsymbol{\xi}(\hat{\boldsymbol{\theta}}_t)$ from (1) minimizes a task-specific cumulative loss $L(\boldsymbol{\xi}(\hat{\boldsymbol{\theta}}))$.

To achieve a specific learning or control task, one needs to select the most suitable measurement $\boldsymbol{O}_t$. Below, we will present three modes of the proposed OCIL framework. It is worth noting that in different applications, adjustments to the configuration of system $\Sigma(\hat{\boldsymbol{\theta}})$ are required according to the task.

**Online SysID:** For a SysID problem, the goal is to identify the dynamics model of a physical system from the state-input trajectory $\boldsymbol{\xi}^o = \{\boldsymbol{x}_{0:T}^o, \boldsymbol{u}_{0:T-1}\}$, where the superscript $o$ denotes the observed trajectory. The trajectory is often generated by persistent excitation of the system without considering any control objectives (Keesman, 2011). Therefore, we can set $J(\boldsymbol{x}_{0:T}, \boldsymbol{u}_{0:T-1}, \hat{\boldsymbol{\theta}}) = 0$:

$$\Sigma(\hat{\boldsymbol{\theta}}) : \begin{array}{ll} \text{dynamics:} & \boldsymbol{x}_{t+1} = \boldsymbol{f}(\boldsymbol{x}_t, \boldsymbol{u}_t, \hat{\boldsymbol{\theta}}), \quad \text{with } \boldsymbol{x}_0 \text{ given,} \\ \text{objective:} & J(\boldsymbol{x}_{0:T}, \boldsymbol{u}_{0:T-1}, \hat{\boldsymbol{\theta}}) = 0. \end{array} \qquad (7)$$

To identify the model dynamics, namely finding the $\boldsymbol{\theta}^*$ in the dynamics $\boldsymbol{f}(\boldsymbol{x}_t, \boldsymbol{u}_t, \boldsymbol{\theta}^*)$, one could design the signed residual function to represent the discrepancy between the observed trajectory and the trajectory produced by $\hat{\boldsymbol{\theta}}$, i.e. $\boldsymbol{l}(\boldsymbol{\xi}_t(\hat{\boldsymbol{\theta}}), \boldsymbol{\xi}_t^o) = \boldsymbol{\xi}_t^o - \boldsymbol{\xi}_t(\hat{\boldsymbol{\theta}})$, where $\boldsymbol{\xi}_t^o$ is a slice of $\boldsymbol{\xi}^o$ at time $t$. In the SysID mode, the measurement $\boldsymbol{O}_t$ received at time $t$ is a slice of the trajectory of a physical system $\boldsymbol{\xi}_t^o$.

**Online Imitation Learning:** The objective function and the model dynamics are parameterized by an unknown $\boldsymbol{\theta}^*$. The OC system follows (3). Suppose one can observe the measurement of the expert demonstration $\boldsymbol{y}_t^*$ at each time $t$. Then, the signed residual function can be designed as $\boldsymbol{l}(\boldsymbol{\xi}_t(\hat{\boldsymbol{\theta}}), \boldsymbol{y}_t^*) = \boldsymbol{y}_t^* - \boldsymbol{g}(\boldsymbol{x}_t(\hat{\boldsymbol{\theta}}), \boldsymbol{u}_t(\hat{\boldsymbol{\theta}}))$. In this case, the measurement $\boldsymbol{O}_t$ received at time $t$ is the expert demonstration $\boldsymbol{y}_t^*$. The optimal demonstration can vary between being continuous or sparse, depending on practical application scenarios.

**Tuning Policy On-the-fly:** For an autonomous system, one would like to obtain a control policy such that the trajectory minimizes certain task loss. This mode considers a feedback controller which is parameterized by $\hat{\boldsymbol{\theta}}$, i.e. $\boldsymbol{u}_t = \boldsymbol{\mu}(\boldsymbol{x}_t, \hat{\boldsymbol{\theta}})$. Then the OC system is written as follows:

$$\Sigma(\hat{\boldsymbol{\theta}}) : \begin{array}{ll} \text{dynamics:} & \boldsymbol{x}_{t+1} = \boldsymbol{f}(\boldsymbol{x}_t, \boldsymbol{\mu}(\boldsymbol{x}_t, \hat{\boldsymbol{\theta}})), \quad \text{with } \boldsymbol{x}_0 \text{ given,} \\ \text{objective:} & J(\boldsymbol{x}_{0:T}, \boldsymbol{u}_{0:T-1}, \hat{\boldsymbol{\theta}}) = \sum_{t=0}^{T-1} c(\boldsymbol{x}_t, \boldsymbol{\mu}(\boldsymbol{x}_t, \hat{\boldsymbol{\theta}})) + h(\boldsymbol{x}_T). \end{array} \qquad (8)$$

Then we can design the signed residual function such that it represents trajectory tracking. For instance, the signed residual function could be $\boldsymbol{l}(\boldsymbol{\xi}_t(\hat{\boldsymbol{\theta}}), \boldsymbol{\xi}_t^d) = \boldsymbol{\xi}_t^d - \boldsymbol{\xi}_t(\hat{\boldsymbol{\theta}})$, where $\boldsymbol{\xi}_t^d$ is a slice of desired trajectory to track at time $t$.

## 3 Main Results

The proposed OCIL consists of two main components, both of which are inspired by control theory. Specifically, OCIL first proposes an online parameter estimator based on the extended Kalman filter (EKF). Going forward, we will show the challenge of obtaining the Kalman gain. To tackle this challenge, the gradient information for the loss with respect to the tunable parameter is required. Therefore, OCIL employs a gradient generator (GG) based on Pontryagin Differential Programming to calculate the exact gradient. Then the proposed OCIL framework will be introduced and supported with theoretical analysis.

### 3.1 Online Parameter Estimator

To minimize the cumulative task loss $L(\boldsymbol{\xi}(\hat{\boldsymbol{\theta}}))$ with measurement $\boldsymbol{O}_t$, which is unavailable until time $t$, the optimization problem that needs to be solved in an online fashion is:

$$\min_{\boldsymbol{\theta}} \quad \sum_{t=0}^{T} \|\boldsymbol{l}(\boldsymbol{\xi}_t(\hat{\boldsymbol{\theta}}), \boldsymbol{O}_t)\|^2 \quad \text{subject to} \quad \boldsymbol{\xi}(\hat{\boldsymbol{\theta}}) \text{ is the trajectory of (3).} \tag{9}$$

The optimization problem (9) is essentially a least squares problem, although under constraints. One of the most famous methods to solve the least squares problems incrementally is the EKF (Bertsekas, 1996; Ribeiro, 2004). The EKF was proposed to incrementally estimate the state of a system using measured output available at each time step. In our problem setting, instead of estimating the state of a system, our goal is to estimate the parameter $\boldsymbol{\theta}^*$ by utilizing the measurement $\boldsymbol{O}_t$ that is available at each time $t$. Therefore, by considering the parameter $\boldsymbol{\theta}^*$ as the state to be estimated, one can introduce a new dynamical system:

$$\text{dynamics: } \boldsymbol{\theta}_{t+1} = \boldsymbol{\theta}_t, \text{ with } \boldsymbol{\theta}_0 = \boldsymbol{\theta}^*, \quad \text{measurement: } \boldsymbol{O}_t = \boldsymbol{h}(\boldsymbol{\xi}_t(\boldsymbol{\theta}_t)) + \boldsymbol{v}_t, \tag{10}$$

The online estimation of $\boldsymbol{\theta}^*$ via EKF can be done as follows (Ribeiro, 2004):

$$\hat{\boldsymbol{\theta}}_t^- := \hat{\boldsymbol{\theta}}_{t-1}, \ \boldsymbol{P}_t^- := \boldsymbol{P}_{t-1} \tag{11a}$$

$$\boldsymbol{K}_t := \boldsymbol{P}_t^- \boldsymbol{L}_t'(\boldsymbol{L}_t \boldsymbol{P}_t^- \boldsymbol{L}_t' + \boldsymbol{R}_t)^{-1}, \ \boldsymbol{P}_t := (\boldsymbol{I}_p - \boldsymbol{K}_t \boldsymbol{L}_t)\boldsymbol{P}_t^-, \ \hat{\boldsymbol{\theta}}_t := \hat{\boldsymbol{\theta}}_t^- + \boldsymbol{K}_t(\boldsymbol{O}_t - \boldsymbol{h}(\boldsymbol{\xi}_t(\hat{\boldsymbol{\theta}}_t^-))), \tag{11b}$$

$$\boldsymbol{L}_t \triangleq \frac{d\boldsymbol{l}(\boldsymbol{\xi}_t(\boldsymbol{\theta}_t), \boldsymbol{O}_t)}{d\boldsymbol{\theta}_t}\Big|_{\boldsymbol{\theta}_t = \hat{\boldsymbol{\theta}}_t^-} \in \mathbb{R}^{r \times p} \tag{12}$$

where (11a) predicts the dynamics; (11b) updates the parameter estimate. Here, the superscript $^-$ means the term is not yet updated by measurement residual; $\boldsymbol{P}_t \in \mathbb{R}^{p \times p}$ is a positive-definite matrix that denotes the covariance of the estimate; $\boldsymbol{K}_t \in \mathbb{R}^{p \times r}$ denotes the Kalman gain. Throughout the estimation process, all of the terms are known except $\boldsymbol{L}_t$. It is challenging to obtain this term as the signed residual function $\boldsymbol{l}(\boldsymbol{\xi}_t(\hat{\boldsymbol{\theta}}), \boldsymbol{O}_t)$ is not an explicit function of $\boldsymbol{\theta}$. In the next subsection, we will present a *gradient generator* which computes the exact value for $\boldsymbol{L}_t$.

### 3.2 Gradient Generator

In this section, for notation simplicity, the parameter estimate $\hat{\boldsymbol{\theta}}_t^-$ is simplified to $\boldsymbol{\theta}$; $\frac{d\boldsymbol{l}(\boldsymbol{\xi}_t(\boldsymbol{\theta}_t), \boldsymbol{O}_t)}{d\boldsymbol{\theta}_t}\Big|_{\boldsymbol{\theta}_t = \hat{\boldsymbol{\theta}}_t^-}$ is written as $\frac{d\boldsymbol{l}(\boldsymbol{\xi}_t(\boldsymbol{\theta}))}{d\boldsymbol{\theta}}$. To obtain the gradient $\frac{d\boldsymbol{l}(\boldsymbol{\xi}_t(\boldsymbol{\theta}))}{d\boldsymbol{\theta}}$, one can employ the chain rule by definition,

$$\frac{d\boldsymbol{l}(\boldsymbol{\xi}_t(\boldsymbol{\theta}))}{d\boldsymbol{\theta}} = \frac{\partial \boldsymbol{l}(\boldsymbol{\xi}_t(\boldsymbol{\theta}))}{\partial \boldsymbol{\xi}_t(\boldsymbol{\theta})} \frac{\partial \boldsymbol{\xi}_t(\boldsymbol{\theta})}{\partial \boldsymbol{\theta}}, \tag{13}$$

where $\frac{\partial \boldsymbol{l}(\boldsymbol{\xi}_t(\boldsymbol{\theta}))}{\partial \boldsymbol{\xi}_t(\boldsymbol{\theta})}$ is known since the signed residual function is pre-designed. The challenge that remains is to find the partial derivative $\frac{\partial \boldsymbol{\xi}_t(\boldsymbol{\theta})}{\partial \boldsymbol{\theta}}$, i.e. an analytical relation between trajectory $\boldsymbol{\xi}_t$ and the tunable parameter $\boldsymbol{\theta}$. To tackle this challenge, the gradient generator in Jin et al. (2020) is used to obtain the exact value of $\frac{\partial \boldsymbol{\xi}_t(\boldsymbol{\theta})}{\partial \boldsymbol{\theta}}$.

Given the OC system (3), one can obtain the Hamiltonian equation

$$H_t = c(\boldsymbol{x}_t, \boldsymbol{u}_t, \boldsymbol{\theta}) + \boldsymbol{f}(\boldsymbol{x}_t, \boldsymbol{u}_t, \boldsymbol{\theta})' \boldsymbol{\lambda}_{t+1} \tag{14}$$

for all $t = 0, \cdots, T-1$, where $\boldsymbol{\lambda}_t \in \mathbb{R}^n$ denotes the Lagrangian multiplier associated with the equality constraint of model dynamics. With the definition of $\boldsymbol{\xi}(\boldsymbol{\theta})$, one has $\frac{\partial \boldsymbol{\xi}(\boldsymbol{\theta})}{\partial \boldsymbol{\theta}} = \text{col}\{\frac{\partial \boldsymbol{x}_{1:T}(\boldsymbol{\theta})}{\partial \boldsymbol{\theta}}, \frac{\partial \boldsymbol{u}_{0:T-1}(\boldsymbol{\theta})}{\partial \boldsymbol{\theta}}\}$. By defining

$$X_t \triangleq \frac{\partial \boldsymbol{x}_t(\boldsymbol{\theta})}{\partial \boldsymbol{\theta}} \in \mathbb{R}^{n \times p}, \quad U_t \triangleq \frac{\partial \boldsymbol{u}_t(\boldsymbol{\theta})}{\partial \boldsymbol{\theta}} \in \mathbb{R}^{m \times p}, \tag{15}$$

one can utilize the following lemma from Jin et al. (2020) to obtain the partial derivatives $\frac{\partial \boldsymbol{\xi}_t(\boldsymbol{\theta})}{\partial \boldsymbol{\theta}}$:

**Lemma 1.** *Jin et al. (2020) Define the Jacobian and Hessian matrices related to $\boldsymbol{\xi}(\boldsymbol{\theta})$ as:*

$$
\begin{aligned}
\boldsymbol{F}_t &= \frac{\partial \boldsymbol{f}}{\partial \boldsymbol{x}_t}, \ \boldsymbol{G}_t = \frac{\partial \boldsymbol{f}}{\partial \boldsymbol{u}_t}, \ \boldsymbol{E}_t = \frac{\partial \boldsymbol{f}}{\partial \boldsymbol{\theta}}, \boldsymbol{H}_t^{xx} = \frac{\partial^2 H_t}{\partial \boldsymbol{x}_t \partial \boldsymbol{x}_t}, \ \boldsymbol{H}_t^{xu} = \frac{\partial^2 H_t}{\partial \boldsymbol{x}_t \partial \boldsymbol{u}_t} = (\boldsymbol{H}_t^{ux})', \\
\boldsymbol{H}_t^{uu} &= \frac{\partial^2 H_t}{\partial \boldsymbol{u}_t \partial \boldsymbol{u}_t}, \boldsymbol{H}_t^{x\theta} = \frac{\partial^2 H_t}{\partial \boldsymbol{x}_t \partial \boldsymbol{\theta}}, \ \boldsymbol{H}_t^{u\theta} = \frac{\partial^2 H_t}{\partial \boldsymbol{u}_t \partial \boldsymbol{\theta}}, \ \boldsymbol{H}_T^{xx} = \frac{\partial^2 h}{\partial \boldsymbol{x}_T \partial \boldsymbol{x}_T}, \ \boldsymbol{H}_T^{x\theta} = \frac{\partial^2 h}{\partial \boldsymbol{x}_T \partial \boldsymbol{\theta}}.
\end{aligned}
\tag{16}
$$

*If $\boldsymbol{H}_t^{uu}$ is invertible for all $t = 0, \cdots, T-1$, the following recursions from $t = T$ to $t = 0$ hold:*

$$
\begin{aligned}
\boldsymbol{V}_t &= \boldsymbol{C}_t + \boldsymbol{A}_t'(\boldsymbol{I} + \boldsymbol{V}_{t+1}\boldsymbol{B}_t)^{-1}\boldsymbol{V}_{t+1}\boldsymbol{A}_t, \\
\boldsymbol{W}_t &= \boldsymbol{A}_t'(\boldsymbol{I} + \boldsymbol{V}_{t+1}\boldsymbol{B}_t)^{-1}(\boldsymbol{W}_{t+1} + \boldsymbol{V}_{t+1}\boldsymbol{M}_t) + \boldsymbol{N}_t,
\end{aligned}
\tag{17}
$$

*with $\boldsymbol{V}_T = \boldsymbol{H}_T^{xx}$ and $\boldsymbol{W}_T = \boldsymbol{H}_T^{x\theta}$. Here, $\boldsymbol{A}_t = \boldsymbol{F}_t - \boldsymbol{G}_t(\boldsymbol{H}_t^{uu})^{-1}\boldsymbol{H}_t^{ux}$, $\boldsymbol{B}_t = \boldsymbol{G}_t(\boldsymbol{H}_t^{uu})^{-1}\boldsymbol{G}_t'$, $\boldsymbol{M}_t = \boldsymbol{E}_t - \boldsymbol{G}_t(H_t^{uu})'\boldsymbol{H}_t^{u\theta}$, $\boldsymbol{C}_t = \boldsymbol{H}_t^{xx} - \boldsymbol{H}_t^{xu}(\boldsymbol{H}_t^{uu})^{-1}\boldsymbol{H}_t^{ux}$, $\boldsymbol{N}_t = \boldsymbol{H}_t^{x\theta} - \boldsymbol{H}_t^{xu}(\boldsymbol{H}_t^{uu})'\boldsymbol{H}_t^{u\theta}$ are all known given (16). Then, the partial derivative $\frac{\partial \boldsymbol{\xi}(\boldsymbol{\theta})}{\partial \boldsymbol{\theta}}$ can be obtained by recursively solving the following equations from $t = 0$ to $T-1$ with $\boldsymbol{X}_0(\boldsymbol{\theta}) = \boldsymbol{0}$:*

$$
\begin{aligned}
\boldsymbol{U}_t &= -(\boldsymbol{H}_t^{uu})^{-1}(\boldsymbol{H}_t^{ux}\boldsymbol{X}_t + \boldsymbol{H}_t^{u\theta} + \boldsymbol{G}_t'(\boldsymbol{I} + \boldsymbol{V}_{t+1}\boldsymbol{B}_t)^{-1}(\boldsymbol{V}_{t+1}\boldsymbol{A}_t\boldsymbol{X}_t + \boldsymbol{V}_{t+1}\boldsymbol{M}_t + \boldsymbol{W}_{t+1})), \\
\boldsymbol{X}_{t+1} &= \boldsymbol{F}_t\boldsymbol{X}_t + \boldsymbol{G}_t\boldsymbol{U}_t + \boldsymbol{E}_t.
\end{aligned}
\tag{18}
$$

The terms in (16) are based on the trajectory $\boldsymbol{\xi}(\boldsymbol{\theta})$ and the associated Lagrangian multiplier $\boldsymbol{\lambda}_{0:T-1}$. According to the discrete-time Pontryagin Maximum Principle (Jin et al., 2020), the trajectory of the Lagrangian multiplier can be obtained by

$$\boldsymbol{\lambda}_T = \frac{\partial h}{\partial \boldsymbol{x}_T}, \quad \boldsymbol{\lambda}_t \triangleq \frac{\partial H_t}{\partial \boldsymbol{x}_t} = \frac{\partial c}{\partial \boldsymbol{x}_t} + \frac{\partial h}{\partial \boldsymbol{x}_t}\boldsymbol{\lambda}_{t+1}, \text{ for } t = T-1, \cdots, 1. \tag{19}$$

**Remark 2.** *Lemma 1 proposes a recursive way to obtain the exact gradient of the trajectory $\boldsymbol{\xi}(\boldsymbol{\theta})$ with respect to the parameter $\boldsymbol{\theta}$, i.e. $\frac{\partial \boldsymbol{\xi}(\boldsymbol{\theta})}{\partial \boldsymbol{\theta}}$.*

### 3.3 OCIL Framework

With the online parameter estimator and the gradient generator, we propose the Online Control-Informed Learning framework in Fig. 1. The framework is summarized in Algorithm 2.

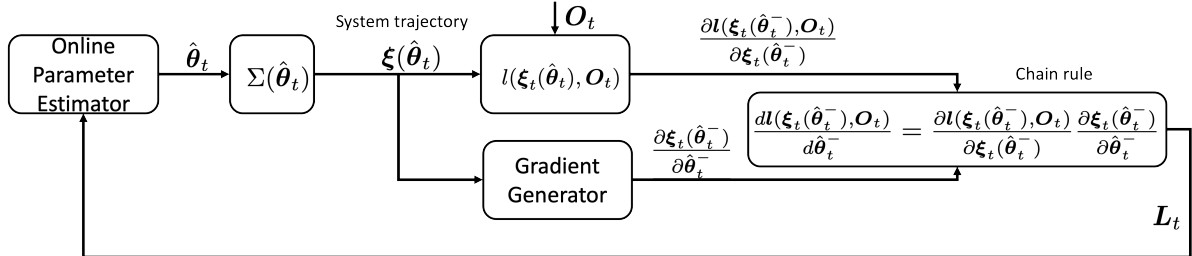

Figure 1: Framework of Online Control-Informed Learning.

As shown in Fig. 1, at each time step, the predefined OC system $\Sigma(\hat{\boldsymbol{\theta}}_t)$ generates a system trajectory $\boldsymbol{\xi}(\hat{\boldsymbol{\theta}}_t)$ by performing optimal control with given $\boldsymbol{x}_0$ and $\hat{\boldsymbol{\theta}}_t$. The trajectory $\boldsymbol{\xi}(\hat{\boldsymbol{\theta}}_t)$ is then fed into the signed residual function $\boldsymbol{l}(\boldsymbol{\xi}_t(\hat{\boldsymbol{\theta}}_t), \boldsymbol{O}_t)$ and the gradient generator. Along with the information $\boldsymbol{O}_t$ obtained at time $t$, the signed residual function generates $\frac{\partial \boldsymbol{l}(\boldsymbol{\xi}_t(\hat{\boldsymbol{\theta}}_t^-), \boldsymbol{O}_t)}{\partial \boldsymbol{\xi}_t(\hat{\boldsymbol{\theta}}_t^-)}$, while $\frac{\partial \boldsymbol{\xi}_t(\hat{\boldsymbol{\theta}}_t^-)}{\partial \hat{\boldsymbol{\theta}}_t^-}$ is generated by the gradient generator in Algorithm 1. The chain rule is then performed to obtain the Jacobian matrix $\boldsymbol{L}_t$, which is then passed into the online parameter estimator for the estimation of $\boldsymbol{\theta}^*$.

---

**Algorithm 1:** Gradient Generator (GG)

**Input:** Trajectory $\boldsymbol{\xi}(\hat{\boldsymbol{\theta}}_t^-)$ from $\Sigma(\hat{\boldsymbol{\theta}}_t^-)$

1 Compute the coefficient matrices in (16) ;
2 Set $\boldsymbol{V}_T = \boldsymbol{H}_T^{xx}$ and $\boldsymbol{W}_T = \boldsymbol{H}_T^{x\theta}$;
3 **for** $t \leftarrow T$ *to* $0$ *by* $\Delta t$ **do**
4 $\quad$ Update $\boldsymbol{V}_t$ and $\boldsymbol{W}_t$ using (17)
5 Set $\boldsymbol{X}_0(\hat{\boldsymbol{\theta}}_t^-) = \boldsymbol{0}$;
6 **for** $t \leftarrow 0$ *to* $T$ *by* $\Delta t$ **do**
7 $\quad$ Update $\boldsymbol{X}_t(\hat{\boldsymbol{\theta}}_t^-)$ and $\boldsymbol{U}_t(\hat{\boldsymbol{\theta}}_t^-)$ using (18)

**Output:** $\frac{\partial \boldsymbol{\xi}(\hat{\boldsymbol{\theta}}_t^-)}{\partial \hat{\boldsymbol{\theta}}_t^-} = \{\boldsymbol{X}_{1:T}(\hat{\boldsymbol{\theta}}_t^-), \boldsymbol{U}_{0:T-1}(\hat{\boldsymbol{\theta}}_t^-)\}$

---

**Algorithm 2:** Online Control-Informed Learning

**System and Residual:** $\Sigma(\hat{\boldsymbol{\theta}})$ and $\boldsymbol{l}(\boldsymbol{\xi}_t(\hat{\boldsymbol{\theta}}), \boldsymbol{O}_t)$
**Initialize:** $\hat{\boldsymbol{\theta}}_0, \boldsymbol{P}_0$

1 **for** $t = t_0, t_1, \cdots$ **do**
2 $\quad$ Obtain new information $\boldsymbol{O}_t$;
3 $\quad$ Solve $\boldsymbol{\xi}(\hat{\boldsymbol{\theta}}_t)$ from current OC system $\Sigma(\hat{\boldsymbol{\theta}}_t)$;
4 $\quad$ Obtain $\frac{\partial \boldsymbol{\xi}_t(\hat{\boldsymbol{\theta}}_t^-)}{\partial \hat{\boldsymbol{\theta}}_t^-}$ with GG in Algorithm 1;
5 $\quad$ Obtain $\frac{\partial \boldsymbol{l}(\boldsymbol{\xi}_t(\hat{\boldsymbol{\theta}}_t^-), \boldsymbol{O}_t)}{\partial \boldsymbol{\xi}_t(\hat{\boldsymbol{\theta}}_t^-)}$ from $\boldsymbol{l}(\boldsymbol{\xi}_t(\hat{\boldsymbol{\theta}}_t^-), \boldsymbol{O}_t)$;
6 $\quad$ Obtain $\boldsymbol{L}_t$ via the chain rule (13);
7 $\quad$ Update $\hat{\boldsymbol{\theta}}_t$ using the estimator (11);

---

## 3.4 Convergence Analysis

This subsection presents the convergence analysis of the online parameter estimator. The analysis employs a candidate Lyapunov function and introduces how the measurement covariance matrix $\boldsymbol{R}_t$ affects the convergence of the cumulative loss $L(\boldsymbol{\xi}(\hat{\boldsymbol{\theta}}))$. In this section, for brevity, the signed residual function $\boldsymbol{l}(\boldsymbol{\xi}_t(\hat{\boldsymbol{\theta}}), \boldsymbol{O}_t)$ is written as $\boldsymbol{l}(\boldsymbol{\xi}_t(\hat{\boldsymbol{\theta}}))$. Suppose for a specific task, the optimal cumulative loss $L(\boldsymbol{\xi}(\boldsymbol{\theta}^*)) = 0$. Then, we define the estimation error as $\tilde{\boldsymbol{\theta}}_t = \boldsymbol{\theta}^* - \hat{\boldsymbol{\theta}}_t$. Furthermore, we define

$$\text{Measurement error: } \boldsymbol{e}_t = \boldsymbol{l}(\boldsymbol{\xi}(\boldsymbol{\theta}^*)) - \boldsymbol{l}(\boldsymbol{\xi}(\hat{\boldsymbol{\theta}}_t^-))$$
$$\text{Prediction error: } \tilde{\boldsymbol{\theta}}_t^- = \boldsymbol{\theta}^* - \hat{\boldsymbol{\theta}}_t^-. \tag{20}$$

To perform the convergence analysis, a candidate Lyapunov function is employed:

$$V_t = \tilde{\boldsymbol{\theta}}_t' \boldsymbol{P}_t^{-1} \tilde{\boldsymbol{\theta}}_t. \tag{21}$$

The goal here is to determine conditions for which the candidate Lyapunov function $\{V_t\}_{t=1,2,\dots}$ is a decreasing sequence, i.e. $V_{t+1} - V_t \leq 0, \forall t$. For rigorous analysis of the candidate Lyapunov function, as proposed in Boutayeb et al. (1997), unknown diagonal matrices $\boldsymbol{\mathcal{F}}_t \in \mathbb{R}^{r \times r}$ and $\boldsymbol{\mathcal{G}}_t \in \mathbb{R}^{p \times p}$ are introduced to model the measurement and prediction error defined in (20):

$$\boldsymbol{\mathcal{F}}_t \boldsymbol{e}_t = \boldsymbol{L}_t \tilde{\boldsymbol{\theta}}_t^-, \quad \tilde{\boldsymbol{\theta}}_t^- = \boldsymbol{\mathcal{G}}_t \tilde{\boldsymbol{\theta}}_{t-1}. \tag{22}$$

To ensure convergence of the proposed estimator, the following assumptions need to be made.

**Assumption 1.** *The derivative $\boldsymbol{L}_t = \frac{d\boldsymbol{l}(\boldsymbol{\xi}_t(\hat{\boldsymbol{\theta}}_t^-))}{d\hat{\boldsymbol{\theta}}_t^-}$ is of full rank for every $\hat{\boldsymbol{\theta}}_t^-$.*

**Remark 3.** *The discrete-time dynamical system (10) satisfies the observability rank condition, i.e., for every $\hat{\boldsymbol{\theta}}_t^-$, $\text{rank}(\text{col}\{\frac{d\boldsymbol{l}(\boldsymbol{\xi}_t(\hat{\boldsymbol{\theta}}_t^-))}{d\hat{\boldsymbol{\theta}}_t^-}, \frac{d\boldsymbol{l}(\boldsymbol{\xi}_t(\hat{\boldsymbol{\theta}}_t^-))}{d\hat{\boldsymbol{\theta}}_t^-} \boldsymbol{I}_p, \cdots, \frac{d\boldsymbol{l}(\boldsymbol{\xi}_t(\hat{\boldsymbol{\theta}}_t^-))}{d\hat{\boldsymbol{\theta}}_t^-} \boldsymbol{I}_p^{p-1}\}) = p$ (Song & Grizzle, 1992). That means if Assumption 1 is satisfied for every $\hat{\boldsymbol{\theta}}_t^-$, the system (10) is observable for every $\hat{\boldsymbol{\theta}}_t^-$. The observability condition assures that $\boldsymbol{P}_t$ is a bounded matrix from above and below (Song & Grizzle, 1992; Boutayeb & Aubry, 1999).*

As common in the EKF analysis, we adopt the following assumption:

**Assumption 2.** *$\boldsymbol{L}_t$ is a uniformly bounded matrix.*

We have the following lemma to show how the measurement covariance matrix $\boldsymbol{R}_t$ affects the convergence of the tunable parameter. The proof can be found in Appendix A.

**Lemma 2.** *Let Assumptions 1 and 2 hold. If the following inequalities are satisfied:*

$$(\boldsymbol{\mathcal{F}}_t - \boldsymbol{I}_s)^2 \leq \boldsymbol{R}_t(\boldsymbol{L}_t \boldsymbol{P}_t^- \boldsymbol{L}_t' + \boldsymbol{R}_t)^{-1}, \tag{23}$$

$$\boldsymbol{\mathcal{G}}_t' \boldsymbol{P}_t^{-1} \boldsymbol{\mathcal{G}}_t - \boldsymbol{P}_t^{-1} \leq 0, \tag{24}$$

*Then the proposed estimator (11), when used as an observer for the system (10), ensures local asymptotic convergence, i.e. $\lim_{t\to\infty} \tilde{\boldsymbol{\theta}}_t = \boldsymbol{0}$.*

**Remark 4.** *Lemma 2 provides sufficient conditions for the convergence of $\hat{\boldsymbol{\theta}}_t$. As the diagonal matrices $\boldsymbol{\mathcal{F}}_t$ and $\boldsymbol{\mathcal{G}}_t$ are unknown, one can design the matrix $\boldsymbol{R}_t$ to satisfy inequalities (23). For example, one can set the matrix $\boldsymbol{R}_t$ to be sufficiently large, i.e. much larger than $\boldsymbol{L}_t \boldsymbol{P}_t^- \boldsymbol{L}_t'$, so that (23) is satisfied, which means the parameter estimator can tolerate arbitrary large initial prediction error. It is worth to note that as long as (23) and (24) are satisfied, $\hat{\boldsymbol{\theta}}_t$ converges to $\boldsymbol{\theta}^*$ and consequently $\boldsymbol{\mathcal{F}}_t$ and $\boldsymbol{\mathcal{G}}_t$ become identity matrix. In the case when there is no measurement noise, i.e. $\boldsymbol{R}_t = \boldsymbol{0}_{s\times s}$, $\boldsymbol{\mathcal{F}}_t$ and $\boldsymbol{\mathcal{G}}_t$ can only be identity matrices to satisfy the inequalities (23) and (24), indicating the convergence of $\hat{\boldsymbol{\theta}}_t$ to $\boldsymbol{\theta}^*$.*

**Remark 5.** *Equation (23) and (24) indicate one of the limitations of the estimator, which is the selection of initial guess. If the initial guess of $\boldsymbol{\theta}^*$ results in $\boldsymbol{\mathcal{F}}_0$ and $\boldsymbol{\mathcal{G}}_0$ that do not satisfy (23) and (24), the value of the Lyapunov function (21) becomes larger, which leads to even larger $\boldsymbol{\mathcal{F}}_t$ and $\boldsymbol{\mathcal{G}}_t$, causing the estimation to diverge.*

We have the following main theorem shows how the measurement covariance matrix $\boldsymbol{R}_t$ affects the convergence of cumulative loss $L(\boldsymbol{\xi}(\hat{\boldsymbol{\theta}}))$ by utilizing the inequalities introduced in Lemma 2. The proof can be found in Appendix B.

**Theorem 1.** *Let Assumptions 1 and 2 hold. If the inequalities in Lemma 2 are met, then estimating $\boldsymbol{\theta}^*$ with the proposed estimator (11) employing the gradient generator in (17)-(18) ensures local asymptotic convergence of the cumulative loss $L$ in (6) to 0 , i.e. $\lim_{t\to\infty} L(\boldsymbol{\xi}(\hat{\boldsymbol{\theta}})) = 0$.*

## 4 Applications to Different Online Learning Modes and Experiments

This section demonstrates the capability of the proposed OCIL framework with its three modes by three applications, Online Imitation Learning, Online System Identification, and Learning Policy on-the-fly. This section includes a performance comparison with some state-of-the-art frameworks for three environments that are summarized in Table 1. Let $\boldsymbol{O}_t^* = \boldsymbol{h}(\boldsymbol{\xi}_t(\boldsymbol{\theta}^*))$ denotes the measurement without noise. The measurement noise is subject to a multivariate Gaussian distribution $\mathcal{N}(\boldsymbol{O}_t^*, \sigma^2 \boldsymbol{I}_r)$.

To highlight the flexibility of OCIL, each experiment includes two phases: 1) online phase, where OCIL keeps learning the unknown parameter while new data comes in before the final time $T$; 2) offline phase, where OCIL keeps learning the parameter given the learned parameter at time $T$ and the entire trajectory obtained from time $t = 0$ to time $T$. For each environment and task, a terminal time $T \in \mathbb{Z}$ is defined to represent a desired time duration where the system shall finish the task.

To unify the data visualization of both online and offline phases, the horizontal axis represents the number of data points, where a vertical red line corresponds to the final time $T$, i.e. the end of the online phase. The number of data points reflects the number of iterations multiplied by the total number of time steps for each iteration. The solid blue curves indicate the online portion of OCIL, whereas the dashed blue curves indicate the offline portion. For every environment and every method, 5 trials are performed given random initial conditions due to the high computational cost for other methods. The computational performance and analysis for OCIL are shown in Section 5 of the Appendix.

**Online Imitation Learning.** The control objective is parameterized as a weighted distance to the goal. Set the signed residual function of imitation learning $l(\boldsymbol{\xi}_t(\hat{\boldsymbol{\theta}}), \boldsymbol{y}_t^*) = \boldsymbol{y}_t^* - \boldsymbol{g}(\boldsymbol{x}_t(\hat{\boldsymbol{\theta}}), \boldsymbol{u}_t(\hat{\boldsymbol{\theta}}))$. The optimal cumulative loss is zero, i.e. $L(\boldsymbol{\xi}(\boldsymbol{\theta}^*)) = 0$, with full knowledge of the parameter. Four existing methods are used for comparisons: (i) inverse KKT (Englert et al., 2017) (ii) neural policy cloning (Bojarski et al., 2016) and (iii) PDP (Jin et al., 2020). These methods don't handle measurement noise well because of their limitations, so we performed the experiments without including measurement noise for these methods. For OCIL, $\sigma = 0.1$ for all of the systems.

Table 1: Experiment Environments

| Systems | Dynamics parameter $\boldsymbol{\theta}_{dyn}$ | Objective parameter $\boldsymbol{\theta}_{obj}$ |
|---|---|---|
| Cartpole | cart mass, pole mass and length | $c(\boldsymbol{x}, \boldsymbol{u}) = \boldsymbol{\theta}_{obj}\|\boldsymbol{x} - \boldsymbol{x}_g\|^2 + \|\boldsymbol{u}\|^2$ |
| 6-DoF Quadrotor | mass, wing length, inertia matrix | |
| 6-DoF Rocket | mass, rocket length, inertia matrix | $h(\boldsymbol{x}) = \boldsymbol{\theta}_{obj}\|\boldsymbol{x} - \boldsymbol{x}_g\|^2$ |

Fig. 2a-2c summarize the comparison result, where OCIL converges faster and obtains lower loss than the other offline methods, in both online and offline phases. The initial loss for each method is different because the learning representation (parameterization) is different. Thus, it is hard to guarantee that an initial neural network has the same loss as another initial parameter vector. Nevertheless, the initial representation of each method is adjusted such that OCIL does not take advantage of good initialization. Fig. 2a-2c validate the effectiveness of OCIL's both online and offline performance, even with measurement noise.

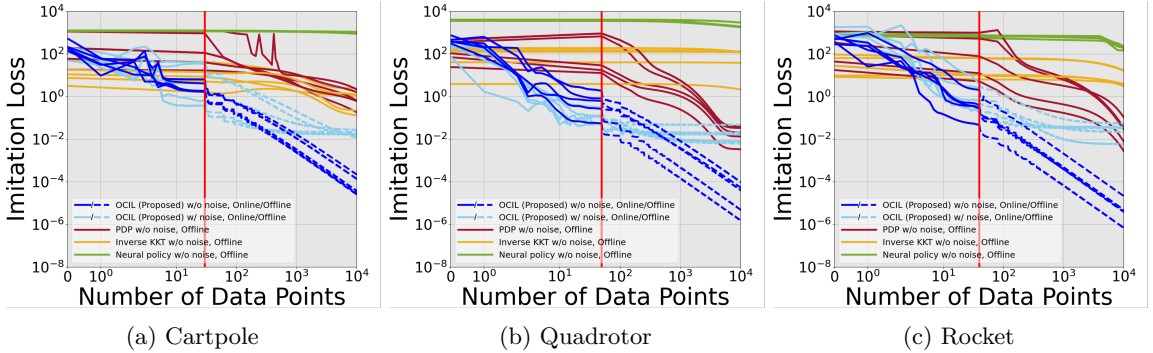

(a) Cartpole        (b) Quadrotor        (c) Rocket

Figure 2: Imitation loss v.s. number of data points

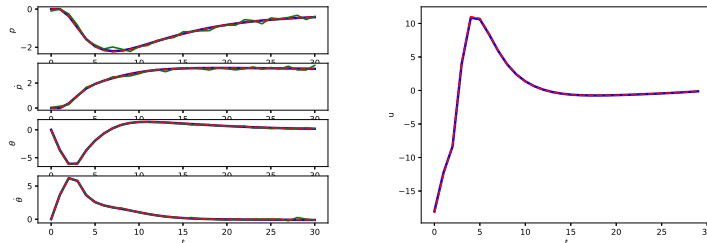

Figure 3: Trajectories of the cartpole in online imitation learning. Blue solid lines: learned trajectory. Green solid lines: observed noisy trajectory. Red dashed lines: ground truth.

**Online System Identification** The signed residual function is set to be $l(\boldsymbol{\xi}_t(\hat{\boldsymbol{\theta}}), \boldsymbol{\xi}_t^o) = \boldsymbol{\xi}_t^o - \boldsymbol{\xi}_t(\hat{\boldsymbol{\theta}})$. The optimal cumulative loss is zero, i.e. $L(\boldsymbol{\xi}(\boldsymbol{\theta}^*)) = 0$, with full knowledge of the parameter. Three other methods are used for comparison: (i) Pytorch Adam solver (Pillonetto et al., 2025), (ii) DMDc (Proctor et al., 2016), and (iii) PDP (Jin et al., 2020). No measurement noise are injected into observed data for existing methods due to their inherent limitations. For OCIL, $\sigma = 0.05$ for all of the systems.

Fig. 5a-5c summarize the result, where OCIL outperforms PDP for faster convergence and lower loss, in both online and offline phases. Different than Online Imitation Learning, OCIL does not decrease its SysID loss significantly at first because the number of data points is not sufficient for online learning. Once the number of data points becomes sufficient, the SysID loss starts decreasing significantly. This phenomenon can also be observed in the other methods, but their critical number of data points is significantly larger than OCIL's. In Fig. 5d-5f, OCIL and other methods are applied to learn the neural dynamics using the same observed

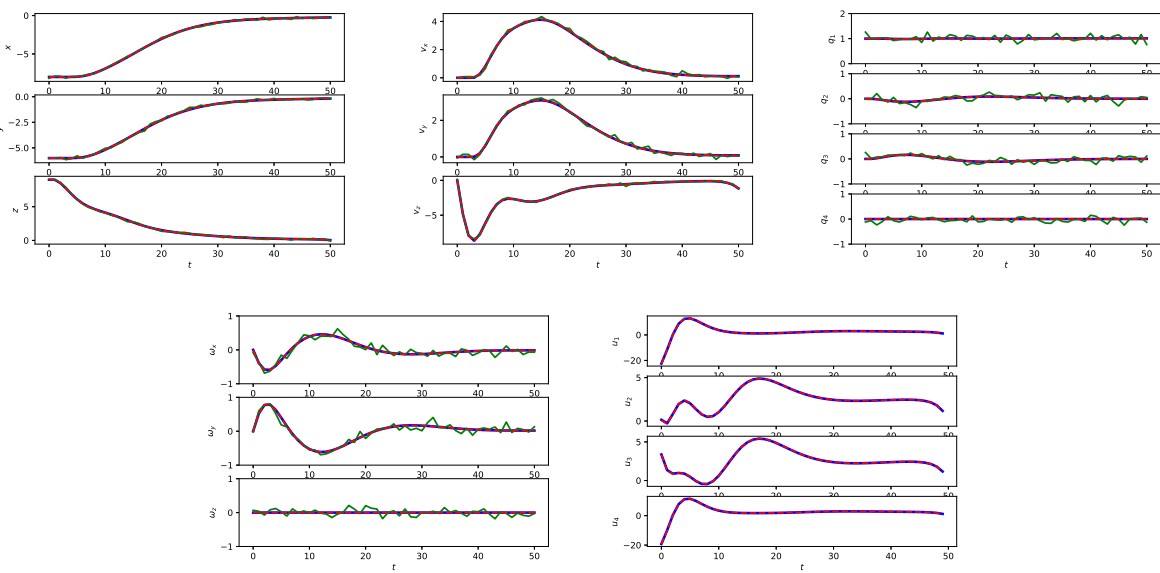

Figure 4: Trajectories of the quadrotor in online imitation learning. Blue solid lines: learned trajectory. Green solid lines: observed noisy trajectory. Red dashed lines: ground truth.

trajectory. It can be seen that OCIL outperforms other methods for lower loss. Fig. 6 demonstrates the capability of OCIL to deal with neural dynamics that have different sizes of NN.

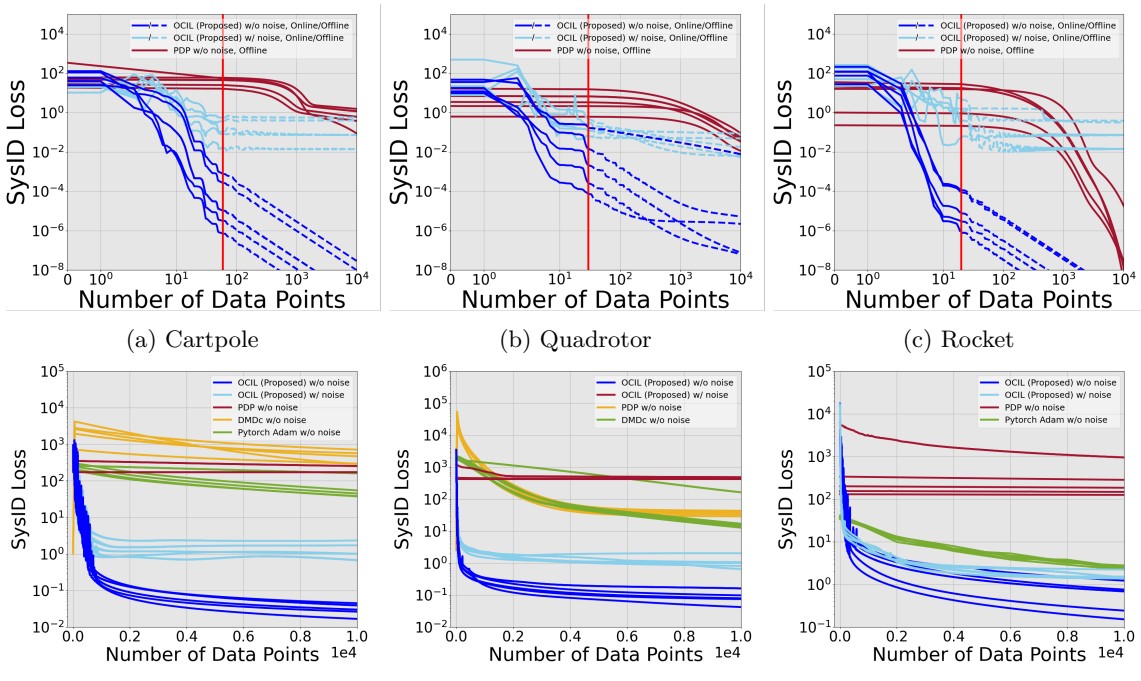

(a) Cartpole       (b) Quadrotor       (c) Rocket

(d) Cartpole, offline, NN dynamics (e) Quadrotor, offline, NN dynamics (f) Rocket, offline, NN dynamics

Figure 5: SysID loss v.s. number of data points

**Policy Tuning On-the-fly.** The parameterized OC system in 8 is used here, where the policy is in a state-feedback form and parameterized by the tunable parameter $\hat{\boldsymbol{\theta}}$. The signed residual function is set to be $l(\boldsymbol{\xi}_t(\hat{\boldsymbol{\theta}}), \boldsymbol{\xi}_t^*) = \boldsymbol{\xi}_t^* - \boldsymbol{\xi}_t(\hat{\boldsymbol{\theta}})$, where $\boldsymbol{\xi}_t^*$ is the trajectory that needs to be tracked. The optimal cumulative loss

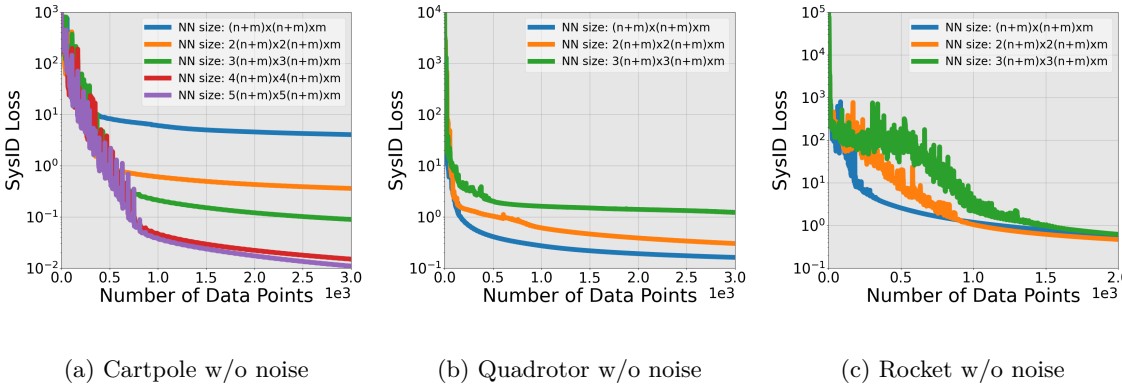

(a) Cartpole w/o noise     (b) Quadrotor w/o noise     (c) Rocket w/o noise

Figure 6: SysID Loss v.s. number of data points, given different sizes of neural dynamics

is zero, i.e. $L(\boldsymbol{\xi}(\boldsymbol{\theta}^*)) = 0$, with full knowledge of the parameter. Other methods are used for comparison (i) iLQR (Li & Todorov, 2004) (ii) GPS (Levine & Abbeel, 2014), and (iii) PDP (Jin et al., 2020). No measurement noise is included for existing methods due to their limitations. For OCIL, $\sigma = 0.1$ for cartpole and quadrotor; $\sigma = 0.25$ for rocket.

Fig. 9a-9c summarize the result, where the loss and its variation of OCIL converge very quickly. The buffers in Fig. 9a-9f indicate 3 times of standard deviation. Fig. 9d-9f presents the online phase of OCIL given 1000 random trials, which further validates the effectiveness and robustness of OCIL given measurement noise.

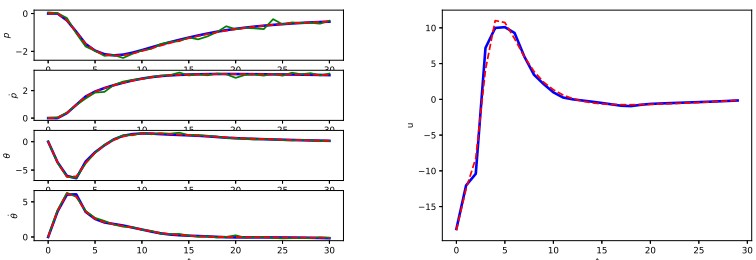

Figure 7: Trajectories of the cartpole in policy tuning on-the-fly. Blue solid lines: learned trajectory. Green solid lines: observed noisy trajectory. Red dashed lines: ground truth.

In general, OCIL from all figures does not have a smooth loss trajectory as the other offline methods. This is because at the online phase, an optimal gain matrix $\boldsymbol{K}_t$ from (11a) is computed to update $\hat{\boldsymbol{\theta}}_t$, whereas the other methods either use a constant or iteration-dependent step size. The optimal gain is conceptually similar to searching an optimal step size in the line-search optimization algorithms. Thus, it is observed that the loss variation, as represented by blue buffers, is relatively high initially but starts decreasing significantly as new data comes in because $\boldsymbol{K}_t$ is continually updated. In contrast, the loss variation barely changes for the other offline methods after some data points.

## 5   Online Computational Performance

The experiments with OCIL were performed on a desktop with one Intel Core i7-8700k CPU with 8GB RAM. No GPU was used. The experiments with other methods were performed on a desktop with one AMD Ryzen 9 5900X CPU, one Nvidia Geforce RTX 4070ti, and 32 GB RAM. A more powerful PC was selected for the other methods because of their high computational cost. As noted at the beginning of Section 4, only 5 trials were conducted due to the computational expense.

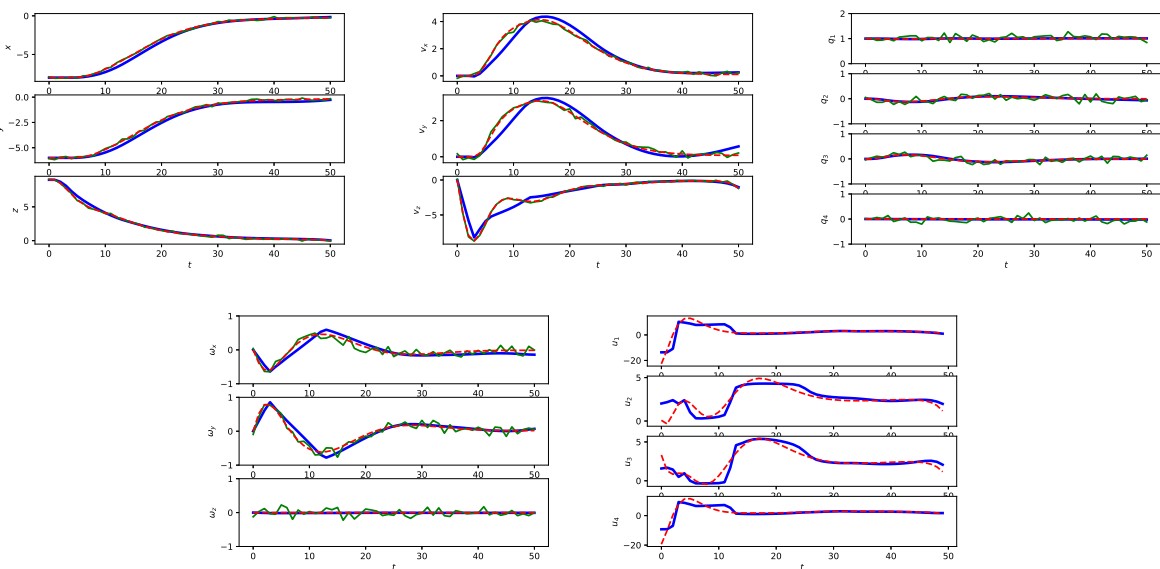

Figure 8: Trajectories of the quadrotor in policy tuning on-the-fly. Blue solid lines: learned trajectory. Green solid lines: observed noisy trajectory. Red dashed lines: ground truth.

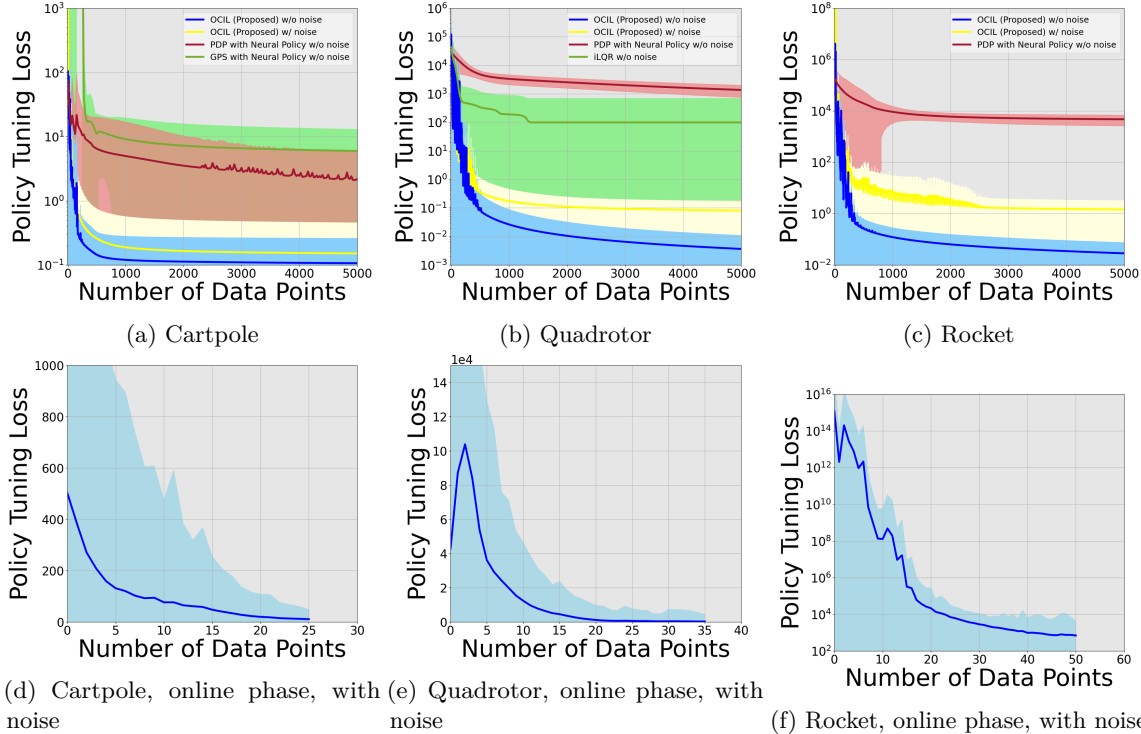

(a) Cartpole     (b) Quadrotor     (c) Rocket

(d) Cartpole, online phase, with noise    (e) Quadrotor, online phase, with noise    (f) Rocket, online phase, with noise

Figure 9: Policy Tuning Loss v.s. number of data points. Buffers represent loss variation under $3\sigma$ with random initial conditions.

To demonstrate that the computational performance of OCIL is enough to be used in an online fashion, we recorded the computational time for OCIL in different modes for 100 trials. The code is implemented in Python, utilizing the CasADi library with the IPOPT solver to solve the OC problem. Table 3 summarizes the OCIL's computational performance for the system identification task for three environments, where OCIL

Time indicates the computational time of running OCIL at each time $t$, i.e. the iteration within the for-loop of Algorithm 2; GG Time indicates the computational time of running gradient generator (GG) at each time $t$, i.e. Algorithm 1; Estimator Time indicates the computational time of updating $\hat{\boldsymbol{\theta}}$, i.e. Line 7 of Algorithm 2; $\Delta$ indicates the time step of each environment, i.e., the time duration between two consecutive data measurements or the maximum allowed time duration of online algorithms to perform computation; Percentage indicates the percentage of the average OCIL time with respect to $\Delta$. The header of Table 4 and Table 2 are the same. Roughly speaking, OCIL time = GG time + Estimator time + Optimal Control computation time.

Table 3 illustrates that OCIL can estimate the dynamical system with neural network representation in an online fashion, within the system frequency of getting new data. Table 4 illustrates that OCIL can tune the neural policy online. As indicated in Line 2 of Algorithm 2, the most computationally heavy part is solving optimal control trajectory in an online fashion, instead of GG and the parameter estimator. As demonstrated at the beginning of this section, OCIL does not require huge computational resources, such as GPU. Therefore, OCIL has the capability to run in an online fashion.

Table 2: Computational Performance for Online Imitation Learning

| Env. | OCIL Time [ms] | GG Time [ms] | Estimator Time [ms] | $\Delta$ [ms] | Percentage |
|---|---|---|---|---|---|
| Cartpole | $62.10 \pm 6.63$ | $7.47 \pm 0.25$ | $0.031 \pm 0.0023$ | 100 | 62.10 % |
| Quadrotor | $81.70 \pm 2.51$ | $21.72 \pm 0.84$ | $0.058 \pm 0.039$ | 100 | 81.70 % |
| Rocket | $72.25 \pm 13.91$ | $19.77 \pm 6.26$ | $0.060 \pm 0.012$ | 100 | 72.25% |

Table 3: Computational Performance for SysID with Neural System

| Env. | OCIL Time [ms] | GG Time [ms] | Estimator Time [ms] | $\Delta$ [ms] | Percentage |
|---|---|---|---|---|---|
| Cartpole | $17.18 \pm 5.15$ | $6.05 \pm 2.59$ | $1.93 \pm 0.85$ | 50 | 34.36 % |
| Quadrotor | $35.53 \pm 8.98$ | $12.18 \pm 4.77$ | $16.18 \pm 6.11$ | 100 | 35.53 % |
| Rocket | $29.54 \pm 8.29$ | $11.25 \pm 4.67$ | $12.89 \pm 5.29$ | 200 | 14.77 % |

Table 4: Computational Performance for Policy Tuning with Neural Policy

| Env. | OCIL Time [ms] | GG Time [ms] | Estimator Time [ms] | $\Delta$ [ms] | Percentage |
|---|---|---|---|---|---|
| Cartpole | $16.41 \pm 5.35$ | $7.72 \pm 3.07$ | $3.96 \pm 1.91$ | 50 | 32.82 % |
| Quadrotor | $62.67 \pm 9.51$ | $30.86 \pm 2.25$ | $22.47 \pm 8.33$ | 100 | 62.67 % |
| Rocket | $59.02 \pm 7.25$ | $33.99 \pm 2.98$ | $12.94 \pm 5.62$ | 100 | 59.02 % |

## 6 Limitations.

This section discusses the major limitations of the proposed framework from three perspectives.

**Local convergence:** Since OCIL is based on first-order gradients, it can only achieve local minima for general non-convex optimal control problems in (3). Furthermore, the general problem proposed in this paper belongs to a bi-level optimization framework. Under certain assumptions such as convexity and smoothness on models (e.g., dynamics model, policy, loss function, and control objective function), global convergence of the bi-level optimization can be established. However, such conditions are too restrictive in the context of dynamic control systems. Therefore, the local convergence analysis based on general nonlinear optimization is enough.

**Parameterization matters for global convergence:** When performing experiments, we find that how models are parameterized matters for good convergence performance. For example, in online SysID mode, we observe that using a neural network dynamics (in Fig. 5d-5f) is more likely to get trapped in local minima than using the true dynamics with unknown parameters (in Fig. 5a-5c). In general, more complex parameterization will bring extreme non-convexity to the optimization problem, making the algorithm more easily trapped in local minima. Determining the parameterization of an object to be learned requires prior or expert knowledge, which is common in ML.

**Initialization matters:** As OCIL borrows how optimal gain updates from EKF, they share the same drawback that convergence depends on the selection of initialization. As shown in Remark 5, a bad initial guess might cause the estimator to diverge according to Lemma 2. Therefore, if a relatively good initial guess is hard to retrieve, one might need to use other methods to cold start OCIL.

## 7 Conclusions

This paper proposes Online Control-Informed Learning (OCIL), an online learning method tailored for diverse learning tasks. By considering an optimal control system with a tunable parameter, OCIL is a unified learning framework that effectively addresses tasks such as online imitation learning, online system identification, and tuning policy on-the-fly. By designing a signed residual function specific to each task and treating the parameter as a state of a new system, we employ the online parameter estimator to estimate the parameter online and minimize the signed residual at each time step. Theoretical analysis establishes the convergence conditions for OCIL, while experiments on various environments, tasks, and existing methods are done to validate its data efficiency, versatility, and robustness against measurement noise.

### Acknowledgments

This material is based upon work supported by the Office of Naval Research (ONR) and Saab, Inc. under the Threat and Situational Understanding of Networked Online Machine Intelligence (TSUNOMI) program (grant no. N00014-23-C-1016). Any opinions, findings and conclusions or recommendations expressed in this material are those of the author(s) and do not necessarily reflect the views of the ONR, the U.S. Government, or Saab, Inc.

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

# Appendix

## A    Proof of Lemma 2

Since the matrices $\boldsymbol{P}_t$ and $\boldsymbol{L}_t$ are bounded according to Assumption 1 and 2, from (11b), one will have:

$$\boldsymbol{K}_t = \boldsymbol{P}_t \boldsymbol{L}_t' \boldsymbol{R}_t^{-1} \tag{25}$$

$$= \boldsymbol{P}_t^- \boldsymbol{L}_t' (\boldsymbol{L}_t \boldsymbol{P}_t^- \boldsymbol{L}_t' + \boldsymbol{R}_t)^{-1}. \tag{26}$$

Then, by taking the inverse of (25) and (26), one will get:

$$\boldsymbol{P}_t^{-1} = (\boldsymbol{P}_t^-)^{-1} + \boldsymbol{L}_t' \boldsymbol{R}_t^{-1} \boldsymbol{L}_t. \tag{27}$$

Substituting (25) into (11b) and subtracting both sides from $\boldsymbol{\theta}_t$, one will have:

$$\tilde{\boldsymbol{\theta}}_t = \tilde{\boldsymbol{\theta}}_t^- - \boldsymbol{P}_t \boldsymbol{L}_t' \boldsymbol{R}_t^{-1} \boldsymbol{e}_t. \tag{28}$$

Then, plug (28) into the Lyapunov function (21):

$$V_t = \tilde{\boldsymbol{\theta}}_t' \boldsymbol{P}_t^{-1} \tilde{\boldsymbol{\theta}}_t \tag{29}$$

$$= (\tilde{\boldsymbol{\theta}}_t^- - \boldsymbol{P}_t \boldsymbol{L}_t' \boldsymbol{R}_t^{-1} \boldsymbol{e}_t)' \boldsymbol{P}_t^{-1} (\tilde{\boldsymbol{\theta}}_t^- - \boldsymbol{P}_t \boldsymbol{L}_t' \boldsymbol{R}_t^{-1} \boldsymbol{e}_t) \tag{30}$$

$$= (\tilde{\boldsymbol{\theta}}_t^-)' \boldsymbol{P}_t^{-1} \tilde{\boldsymbol{\theta}}_t^- - (\tilde{\boldsymbol{\theta}}_t^-)' \boldsymbol{L}_t' \boldsymbol{R}_t^{-1} \boldsymbol{e}_t - \boldsymbol{e}_t' \boldsymbol{R}_t^{-1} \boldsymbol{L}_t \tilde{\boldsymbol{\theta}}_t^- + \boldsymbol{e}_t' \boldsymbol{R}_t^{-1} \boldsymbol{L}_t \boldsymbol{P}_t \boldsymbol{L}_t' \boldsymbol{R}_t^{-1} \boldsymbol{e}_t \tag{31}$$

Next, we plug (27) into (31):

$$V_t = (\tilde{\boldsymbol{\theta}}_t^-)'((\boldsymbol{P}_t^-)^{-1} + \boldsymbol{L}_t' \boldsymbol{R}_t^{-1} \boldsymbol{L}_t) \tilde{\boldsymbol{\theta}}_t^- - (\tilde{\boldsymbol{\theta}}_t^-)' \boldsymbol{L}_t' \boldsymbol{R}_t^{-1} \boldsymbol{e}_t - \boldsymbol{e}_t' \boldsymbol{R}_t^{-1} \boldsymbol{L}_t \tilde{\boldsymbol{\theta}}_t^- + \boldsymbol{e}_t' \boldsymbol{R}_t^{-1} \boldsymbol{L}_t \boldsymbol{P}_t \boldsymbol{L}_t' \boldsymbol{R}_t^{-1} \boldsymbol{e}_t \tag{32}$$

$$= V_t^- + (\tilde{\boldsymbol{\theta}}_t^-)' \boldsymbol{L}_t' \boldsymbol{R}_t^{-1} \boldsymbol{L}_t \tilde{\boldsymbol{\theta}}_t^- - (\tilde{\boldsymbol{\theta}}_t^-)' \boldsymbol{L}_t' \boldsymbol{R}_t^{-1} \boldsymbol{e}_t - \boldsymbol{e}_t' \boldsymbol{R}_t^{-1} \boldsymbol{L}_t \tilde{\boldsymbol{\theta}}_t^- + \boldsymbol{e}_t' \boldsymbol{R}_t^{-1} \boldsymbol{L}_t \boldsymbol{P}_t \boldsymbol{L}_t' \boldsymbol{R}_t^{-1} \boldsymbol{e}_t, \tag{33}$$

where

$$V_t^- = (\tilde{\boldsymbol{\theta}}_t^-)' (\boldsymbol{P}_t^-)^{-1} \tilde{\boldsymbol{\theta}}_t^- \tag{34}$$

$$= (\tilde{\boldsymbol{\theta}}_{t-1})' \boldsymbol{\mathcal{G}}_t' \boldsymbol{P}_{t-1}^{-1} \boldsymbol{\mathcal{G}}_t \tilde{\boldsymbol{\theta}}_{t-1}. \tag{35}$$

Using (22), (33) becomes:

$$V_t = V_t^- + (\tilde{\boldsymbol{\theta}}_t^-)' \boldsymbol{L}_t' \boldsymbol{R}_t^{-1} \boldsymbol{L}_t \tilde{\boldsymbol{\theta}}_t^- - (\tilde{\boldsymbol{\theta}}_t^-)' \boldsymbol{L}_t' \boldsymbol{R}_t^{-1} \boldsymbol{e}_t - \boldsymbol{e}_t' \boldsymbol{R}_t^{-1} \boldsymbol{L}_t \tilde{\boldsymbol{\theta}}_t^- + \boldsymbol{e}_t' \boldsymbol{R}_t^{-1} \boldsymbol{L}_t \boldsymbol{P}_t \boldsymbol{L}_t' \boldsymbol{R}_t^{-1} \boldsymbol{e}_t \tag{36}$$

$$= V_t^- + \boldsymbol{e}_t' \boldsymbol{\mathcal{F}}_t \boldsymbol{R}_t^{-1} \boldsymbol{\mathcal{F}}_t \boldsymbol{e}_t - \boldsymbol{e}_t' \boldsymbol{\mathcal{F}}_t \boldsymbol{R}_t^{-1} \boldsymbol{e}_t - \boldsymbol{e}_t' \boldsymbol{R}_t^{-1} \boldsymbol{\mathcal{F}}_t \boldsymbol{e}_t + \boldsymbol{e}_t' \boldsymbol{R}_t^{-1} \boldsymbol{L}_t \boldsymbol{P}_t \boldsymbol{L}_t' \boldsymbol{R}_t^{-1} \boldsymbol{e}_t \tag{37}$$

$$= V_t^- + \boldsymbol{e}_t' (\boldsymbol{\mathcal{F}}_t \boldsymbol{R}_t^{-1} \boldsymbol{\mathcal{F}}_t - \boldsymbol{\mathcal{F}}_t \boldsymbol{R}_t^{-1} - \boldsymbol{R}_t^{-1} \boldsymbol{\mathcal{F}}_t + \boldsymbol{R}_t^{-1} \boldsymbol{L}_t \boldsymbol{P}_t \boldsymbol{L}_t' \boldsymbol{R}_t^{-1}) \boldsymbol{e}_t. \tag{38}$$

To ensure that the Lyapunov function $\{V_t\}_{t=1,2,\dots}$ is a decreasing sequence, $V_t - V_{t-1} \leq 0$.

$$V_t - V_{t-1} \tag{39}$$

$$= \boldsymbol{e}_t' (\boldsymbol{\mathcal{F}}_t \boldsymbol{R}_t^{-1} \boldsymbol{\mathcal{F}}_t - \boldsymbol{\mathcal{F}}_t \boldsymbol{R}_t^{-1} - \boldsymbol{R}_t^{-1} \boldsymbol{\mathcal{F}}_t + \boldsymbol{R}_t^{-1} \boldsymbol{L}_t \boldsymbol{P}_t \boldsymbol{L}_t' \boldsymbol{R}_t^{-1}) \boldsymbol{e}_t \tag{40}$$

$$+ (\tilde{\boldsymbol{\theta}}_{t-1})' (\boldsymbol{\mathcal{G}}_t' \boldsymbol{P}_{t-1}^{-1} \boldsymbol{\mathcal{G}}_t - \boldsymbol{P}_{t-1}^{-1}) \tilde{\boldsymbol{\theta}}_{t-1} \leq 0. \tag{41}$$

Therefore, to ensure the Lyapunov function is a decreasing sequence,

$$\boldsymbol{\mathcal{F}}_t \boldsymbol{R}_t^{-1} \boldsymbol{\mathcal{F}}_t - \boldsymbol{\mathcal{F}}_t \boldsymbol{R}_t^{-1} - \boldsymbol{R}_t^{-1} \boldsymbol{\mathcal{F}}_t + \boldsymbol{R}_t^{-1} \boldsymbol{L}_t \boldsymbol{P}_t \boldsymbol{L}_t' \boldsymbol{R}_t^{-1} \leq 0, \tag{42}$$

and

$$\boldsymbol{\mathcal{G}}_t' \boldsymbol{P}_{t-1}^{-1} \boldsymbol{\mathcal{G}}_t - \boldsymbol{P}_{t-1}^{-1} \leq 0. \tag{43}$$

With some manipulations:

$$(\boldsymbol{\mathcal{F}}_t - \boldsymbol{I}_s) \boldsymbol{R}_t^{-1} (\boldsymbol{\mathcal{F}}_t - \boldsymbol{I}_s) - \boldsymbol{R}_t^{-1} + \boldsymbol{R}_t^{-1} \boldsymbol{L}_t \boldsymbol{P}_t \boldsymbol{L}_t' \boldsymbol{R}_t^{-1} \leq 0, \tag{44}$$

$$(\boldsymbol{\mathcal{F}}_t - \boldsymbol{I}_s) \boldsymbol{R}_t^{-1} (\boldsymbol{\mathcal{F}}_t - \boldsymbol{I}_s) - \boldsymbol{R}_t^{-1} (\boldsymbol{I}_s - \boldsymbol{L}_t \boldsymbol{P}_t^- \boldsymbol{L}_t' (\boldsymbol{L}_t \boldsymbol{P}_t^- \boldsymbol{L}_t' + \boldsymbol{R}_t)^{-1}) \leq 0. \tag{45}$$

By letting $\boldsymbol{I}_s = (\boldsymbol{L}_t \boldsymbol{P}_t^- \boldsymbol{L}_t' + \boldsymbol{R}_t)(\boldsymbol{L}_t \boldsymbol{P}_t^- \boldsymbol{L}_t' + \boldsymbol{R}_t)^{-1}$, we have

$$(\boldsymbol{\mathcal{F}}_t - \boldsymbol{I}_s)\boldsymbol{R}_t^{-1}(\boldsymbol{\mathcal{F}}_t - \boldsymbol{I}_s) - (\boldsymbol{L}_t \boldsymbol{P}_t^- \boldsymbol{L}_t' + \boldsymbol{R}_t)^{-1} \le 0. \tag{46}$$

Since $\boldsymbol{\mathcal{F}}_t$ and $\boldsymbol{R}_t$ are diagonal matrices, we will have

$$\boldsymbol{R}_t^{-1}(\boldsymbol{\mathcal{F}}_t - \boldsymbol{I}_s)^2 - (\boldsymbol{L}_t \boldsymbol{P}_t^- \boldsymbol{L}_t' + \boldsymbol{R}_t)^{-1} \le 0, \tag{47}$$

which at the end yields:

$$(\boldsymbol{\mathcal{F}}_t - \boldsymbol{I}_s)^2 \le \boldsymbol{R}_t(\boldsymbol{L}_t \boldsymbol{P}_t^- \boldsymbol{L}_t' + \boldsymbol{R}_t)^{-1}, \tag{48}$$

therefore the proof is completed.

## B   Proof of Theorem 1

This proof is straightforward once Lemma 2 is provided. Consider the assumptions 1 and 2 are met, according to Lemma 2, with the exact gradient generated by the gradient generator in (17)-(18), $\lim_{t \to \infty} \tilde{\boldsymbol{\theta}}_t = 0$. As the estimated $\hat{\boldsymbol{\theta}}_t$ converges to the true $\boldsymbol{\theta}^*$, where the true parameter gives zero cumulative loss, the cumulative loss $L(\boldsymbol{\xi}(\hat{\boldsymbol{\theta}}))$ goes to 0.

## C   Experiment Details

### C.1   System/Environment Setups

**Cartpole.** We consider the following continuous dynamics of the cartpole

$$\begin{bmatrix} \dot{p} \\ \ddot{p} \\ \dot{\theta} \\ \ddot{\theta} \end{bmatrix} = \begin{bmatrix} \dot{p} \\ (F + \frac{m_p l \dot{\theta}^2 \sin(\theta)}{m_t}) - \frac{m_p l \ddot{\theta} \cos(\theta)}{m_t} \\ \dot{\theta} \\ \frac{g \sin(\theta) - \cos(\theta)(F + \frac{m_p l \dot{\theta}^2 \sin(\theta)}{m_t})}{l(\frac{4}{3} - \frac{m_p \cos(\theta)^2}{m_t})} \end{bmatrix}, \tag{49}$$

where $p \in \mathbb{R}$ is the horizontal displacement of the cart; $\theta \in \mathbb{R}$ is the pole angle; $F \in \mathbb{R}$ denotes the horizontal force applied to the cart which is between $-1$ and $+1$; $l \in \mathbb{R}$ is the length of the pole; $m_p, m_t \in \mathbb{R}$ are the masses of the pole and total cartpole, respectively. By defining the states and control inputs of the cartpole

$$\boldsymbol{x} \triangleq \begin{bmatrix} p & \dot{p} & \theta & \dot{\theta} \end{bmatrix}' \quad \text{and} \quad \boldsymbol{u} \triangleq F \tag{50}$$

respectively.

**Quadrotor UAV.** We consider a quadrotor UAV with the following dynamics

$$\begin{aligned} \dot{\boldsymbol{p}}_I &= \boldsymbol{v}_I, \\ m\dot{\boldsymbol{v}}_I &= m\boldsymbol{g}_I + \mathbf{F}_I, \\ \dot{\boldsymbol{q}}_{B/I} &= \frac{1}{2}\boldsymbol{\Omega}(\boldsymbol{\omega}_B)\boldsymbol{q}_{B/I}, \\ J_B\dot{\boldsymbol{\omega}}_B &= \mathbf{M}_B - \boldsymbol{\omega} \times J_B\boldsymbol{\omega}_B. \end{aligned} \tag{51}$$

Here, the subscription $_B$ and $_I$ denote a quantity expressed in the body frame and inertial (world) frame, respectively; $m$ and $J_B \in \mathbb{R}^{3 \times 3}$ are the mass and moment of inertia with respect to the body frame of the UAV, respectively. $g$ is the gravitational constant ($g = 10$ m/s$^2$), $\boldsymbol{g}_I = [0, 0, g]'$. $\boldsymbol{p} \in \mathbb{R}^3$ and $\boldsymbol{v} \in \mathbb{R}^3$ are the position and velocity vector of the UAV; $\boldsymbol{\omega}_B \in \mathbb{R}^3$ is the angular velocity vector of the UAV; $\boldsymbol{q}_{B/I} \in \mathbb{R}^4$ is the unit quaternion Kuipers (1999) that describes the attitude of the UAV with respect to the inertial frame; $\boldsymbol{\Omega}(\boldsymbol{\omega}_B)$ is defined as:

$$\boldsymbol{\Omega}(\boldsymbol{\omega}_B) = \begin{bmatrix} 0 & -\omega_x & -\omega_y & -\omega_z \\ \omega_x & 0 & \omega_z & -\omega_y \\ \omega_y & -\omega_z & 0 & \omega_x \\ \omega_z & \omega_y & -\omega_x & 0 \end{bmatrix}, \tag{52}$$

$\mathbf{M}_B \in \mathbb{R}^3$ is the torque applied to the UAV; $\mathbf{F}_I \in \mathbb{R}^3$ is the force vector applied to the UAV center of mass. The total force magnitude $f = \|\mathbf{F}_I\| \in \mathbb{R}$ (along z-axis of the body frame) and torque $\mathbf{M}_B = [M_x, M_y, M_z]'$ are generated by thrust from four rotating propellers $[T_1, T_2, T_3, T_4]'$, their relationship can be expressed as:

$$
\begin{bmatrix} f \\ M_x \\ M_y \\ M_z \end{bmatrix} = \begin{bmatrix} 1 & 1 & 1 & 1 \\ 0 & -l_w/2 & 0 & l_w/2 \\ -l_w/2 & 0 & l_w/2 & 0 \\ c & -c & c & -c \end{bmatrix} \begin{bmatrix} T_1 \\ T_2 \\ T_3 \\ T_4 \end{bmatrix},
\tag{53}
$$

where $l_w$ is the wing length of the UAV and $c$ is a fixed constant. The state and input vectors of the UAV are defined as:

$$
\begin{aligned}
\boldsymbol{x} &\triangleq \begin{bmatrix} \boldsymbol{p}' & \boldsymbol{v}' & \boldsymbol{q}' & \boldsymbol{\omega}' \end{bmatrix}' \in \mathbb{R}^{13}, \\
\boldsymbol{u} &\triangleq \begin{bmatrix} T_1 & T_2 & T_3 & T_4 \end{bmatrix}' \in \mathbb{R}^4.
\end{aligned}
\tag{54}
$$

**Rocket.** The rocket is treated as a rigid body subject to constant gravitational acceleration, $g_I \in \mathbb{R}^3$, and neglects aerodynamic forces. The vehicle is assumed to actuate a single gimbaled rocket engine to generate a thrust vector within a feasible range of magnitudes and gimbal angles. We assume that at the landing phase, the depletion of fuel is insignificant. Therefore, we omit the dynamics of rocket mass. The rocket has the following dynamics:

$$
\begin{aligned}
\dot{\boldsymbol{p}}_\mathcal{I} &= \boldsymbol{v}_\mathcal{I}, \\
\dot{\boldsymbol{v}}_\mathcal{I} &= \frac{1}{m} C_{\mathcal{I}/\mathcal{B}} \mathbf{T}_\mathcal{B} + \mathbf{g}_\mathcal{I}, \\
\dot{\boldsymbol{q}}_{\mathcal{B}/\mathcal{I}} &= \frac{1}{2} \Omega(\boldsymbol{\omega}_\mathcal{B}) \boldsymbol{q}_{\mathcal{B}/\mathcal{I}}, \\
J_\mathcal{B} \dot{\boldsymbol{\omega}}_\mathcal{B} &= \mathbf{M}_\mathcal{B} - [\boldsymbol{\omega}_\mathcal{B} \times] J_\mathcal{B} \boldsymbol{\omega}_\mathcal{B}.
\end{aligned}
\tag{55}
$$

Here, the subscription $_\mathcal{B}$ and $_\mathcal{I}$ denote a quantity expressed in the body frame and inertial (world) frame, respectively; $m$ and $J_\mathcal{B} \in \mathbb{R}^{3 \times 3}$ are the mass and moment of inertia with respect to body frame of the rocket, respectively. $\boldsymbol{p} \in \mathbb{R}^3$ and $\boldsymbol{v} \in \mathbb{R}^3$ are the position and velocity vector of the rocket; $\boldsymbol{\omega}_\mathcal{B} \in \mathbb{R}^3$ is the angular velocity vector of the rocket; $\boldsymbol{q}_{\mathcal{B}/\mathcal{I}} = [q_0, q_1, q_2, q_3]$ is the unit quaternion that describes the attitude of rocket with respect to the inertial frame; $\mathbf{T}_\mathcal{B} \in \mathbb{R}^3$ is the commanded thrust vector; $\mathbf{M}_\mathcal{B} \in \mathbb{R}^3$ is the torque applied to the rocket; $C_{\mathcal{B}/\mathcal{I}}$ is the direction cosine matrix that encodes the attitude transformation from body frame to inertia frame and related to $\boldsymbol{q}_{\mathcal{B}/\mathcal{I}}$ by the following relationship:

$$
C_{\mathcal{B}/\mathcal{I}} = \begin{bmatrix} 1 - 2(q_2^2 + q_3^2) & 2(q_1 q_2 + q_0 q_3) & 2(q_1 q_3 - q_0 q_2) \\ 2(q_1 q_2 - q_0 q_3) & 1 - 2(q_1^2 + q_3^2) & 2(q_2 q_3 + q_0 q_1) \\ 2(q_1 q_3 + q_0 q_2) & 2(q_2 q_3 - q_0 q_1) & 1 - 2(q_1^2 + q_1^2) \end{bmatrix},
$$

The inverse transformation is denoted as $C_{\mathcal{I}/\mathcal{B}} = C_{\mathcal{B}/\mathcal{I}}^T$;

The skew-symmetric matrices $[\boldsymbol{\omega}_\mathcal{B} \times]$ and $\Omega(\boldsymbol{\omega}_\mathcal{B})$ are defined as follow:

$$
[\boldsymbol{\omega}_\mathcal{B} \times] \triangleq \begin{bmatrix} 0 & -\omega_z & \omega_y \\ \omega_z & 0 & -\omega_x \\ -\omega_y & \omega_x & 0 \end{bmatrix}, \quad \Omega(\boldsymbol{\omega}_\mathcal{B}) \triangleq \begin{bmatrix} 0 & -\omega_x & -\omega_y & -\omega_z \\ \omega_x & 0 & \omega_z & -\omega_y \\ \omega_y & -\omega_z & 0 & \omega_x \\ \omega_z & \omega_y & -\omega_x & 0 \end{bmatrix},
$$

The state and input vectors of the rocket are defined as:

$$
\begin{aligned}
\boldsymbol{x} &= \begin{bmatrix} \boldsymbol{p}'_\mathcal{I} & \boldsymbol{v}'_\mathcal{I} & \boldsymbol{q}'_{\mathcal{B}/\mathcal{I}} & \boldsymbol{\omega}'_\mathcal{B} \end{bmatrix}' \in \mathbb{R}^{13}, \\
\boldsymbol{u} &= \mathbf{T}_\mathcal{B} = \begin{bmatrix} T_x & T_y & T_z \end{bmatrix}' \in \mathbb{R}^3,
\end{aligned}
\tag{56}
$$

**Discretization.** Discretization is done by the following discrete-time form

$$
\boldsymbol{x}_{t+1} \approx \boldsymbol{x}_t + \Delta \cdot \boldsymbol{g}(\boldsymbol{x}_t, \boldsymbol{u}_t) \triangleq \boldsymbol{f}(\boldsymbol{x}_t, \boldsymbol{u}_t),
\tag{57}
$$

where $\Delta$ is the discretization interval.

## C.2 Online Imitation Learning

**Data acquisition.** The dataset of expert demonstrations is generated by solving an optimal control system with the true dynamics and control objective parameter $\boldsymbol{\theta}^* = \{\boldsymbol{\theta}_{dyn}, \boldsymbol{\theta}_{obj}\}$ given. We generate five trajectories with different initial conditions $\boldsymbol{x}_0$ and time horizons $T$.

**PDP.** We employed the PDP in Jin et al. (2020) to solve this problem. The learning rate is $\eta = 10^{-4}$. Five trials were run given random initial $\boldsymbol{\theta}_0$.

**Inverse KKT method.** We choose the inverse KKT method Englert et al. (2017) for comparison because it is suitable for learning objective functions for high-dimensional continuous-space systems. We adopt the inverse KKT method and define the KKT loss as the norm-2 violation of the KKT condition by the demonstration data:

$$\min_{\boldsymbol{\theta}, \boldsymbol{\lambda}_{1:T}} \left( \left\| \frac{\partial L}{\partial \boldsymbol{x}_{0:T}}(\boldsymbol{x}_{0:T}^*, \boldsymbol{u}_{0:T-1}^*) \right\|^2 + \left\| \frac{\partial L}{\partial \boldsymbol{u}_{0:T-1}}(\boldsymbol{x}_{0:T}^*, \boldsymbol{u}_{0:T-1}^*) \right\|^2 \right).$$

**Neural policy cloning.** For the neural policy cloning, we directly learn a neural network policy $\boldsymbol{u} = \boldsymbol{\mu}(\boldsymbol{x}, \boldsymbol{\theta})$ from the dataset using supervised learning, that is

$$\min_{\boldsymbol{\theta}} \sum_{t=0}^{T-1} \|\boldsymbol{u}_t^* - \boldsymbol{\mu}(\boldsymbol{x}_t^*, \boldsymbol{\theta})\|^2 \tag{58}$$

## C.3 Online System Identification

**Data acquisition.** In the system identification experiment, we collect a total number of five trajectories from systems with dynamics known, wherein different trajectories $\boldsymbol{\xi}^o = \{\boldsymbol{x}_{0:T}^o, \boldsymbol{u}_{0:T-1}\}$ have different initial conditions $\boldsymbol{x}_0$ and horizons $T$ ($T$ ranges from 10 to 20 depending on different environment and task), with random inputs $\boldsymbol{u}_{0:T-1}$ drawn from uniform distribution.

**PDP.** We employed the PDP in Jin et al. (2020) to solve this problem. The learning rate is $\eta = 10^{-4}$. Five trials were run given random initial $\boldsymbol{\theta}_0$. For the neural dynamics case, the learning rate is $\eta = 10^{-5}$.

**Pytorch Adam to learn neural dynamics.** We consider the dynamics of each system (cartpole, quadrotor, and rocket) are represented by a fully-connected feed-forward neural network $\hat{\boldsymbol{f}}(\boldsymbol{x}_t, \boldsymbol{u}_t, \boldsymbol{\theta})$. The neural network has a layer structure of $(n+m)$-$2(n+m)$-$n$ with $tanh$ activation functions, i.e., there is an input layer with $(n+m)$ neurons equal to the dimension of state, one hidden layer with $2(n+m)$ neurons and one output layer with $n$ neurons. The $\boldsymbol{\xi}^o = \{\boldsymbol{x}_{0:T}^o, \boldsymbol{u}_{0:T-1}\}$ obtained previously are used in stage loss. We conducted five trials for each method with different initial $\boldsymbol{\theta}$. We use Pytorch Adam to train the neural network by minimizing the following residual

$$\min_{\boldsymbol{\theta}} \sum_{t=0}^{T-1} \|\boldsymbol{x}_{t+1}^o - \hat{\boldsymbol{f}}(\boldsymbol{x}_t^o, \boldsymbol{u}_t, \boldsymbol{\theta})\|^2. \tag{59}$$

**DMDc.** The DMDc method Proctor et al. (2016) is a method that is based on Koopman theory to represent nonlinear dynamics with linear dynamics of observables. Observables $\boldsymbol{\psi}(\boldsymbol{x}_t)$ are some basis functions of states. The observable space has a much higher dimension compared to state space. The estimation of the dynamics is achieved by the following optimization:

$$\min_{\boldsymbol{A}, \boldsymbol{B}} \sum_{t=0}^{T-1} \|\boldsymbol{\psi}(\boldsymbol{x}_{t+1}^o) - \boldsymbol{A}\boldsymbol{\psi}(\boldsymbol{x}_t^o) - \boldsymbol{B}\boldsymbol{u}_t\|^2. \tag{60}$$

## C.4 Policy Tuning On-the-fly

**Neural State Feedback Policy.** In this application, we learn the parameters of a neural state feedback policy by minimizing given control objective functions. Specifically, we use a fully connected feed-forward

neural network that has a layer structure of $3n$-$3n$-$m$ with $tanh$ activation functions, i.e., there is an input layer with $3n$ neurons equal to the dimension of state, one hidden layer with $3n$ neurons and one output layer with $m$ neurons. The policy parameter $\boldsymbol{\theta}$ is the neural network parameter. For comparison, we apply the guided policy search (GPS) method Levine & Abbeel (2014) and iLQR Li & Todorov (2004) to solve the same problem.

**PDP.** We employed the PDP in Jin et al. (2020) to solve this problem. The learning rate is set to be $\eta = 10^{-4}$ or $= 10^{-6}$. Five trials were run given random initial $\boldsymbol{\theta}_0$. For the neural objective function case, the learning rate is $\eta = 10^{-5}$.

