# OpenReview forum: "Online Control-Informed Learning"
_TMLR — Accepted by TMLR_

### Review · Reviewer_sujy · 2024-11-18

**Summary Of Contributions:**

This manuscript aims to propose a general learning control framework, named "Online Control-Informed Learning" (OCIL). At its core, the framework combines an Extended Kalman Filter (EKF) with an unusual observation equation capturing the "loss" of a learning problem, with a gradient estimation technique from prior work, needed in the EKF update. The framework is then applied to three learning control use cases in numerical examples.  While such a unified framework might carry some benefits, this reviewer unfortunately failed to see a clear motivation, as well as any technical contribution beyond combining existing techniques and results.

**Audience:**

No

**Claims And Evidence:**

No

**Requested Changes:**

Unfortunately, the technical contributions are unclear to me.

In addition, I have concerns about the presentation, empirical results, discussion of related work, as pointed out above (weaknesses).

In my evaluation, this manuscript needs a substantial revision. Even if the presentation is improved, I have doubts about any clear contribution to the machine learning and control community.

**Strengths And Weaknesses:**

## Strengths

(S1) Ambition to propose a unifying framework capturing different learning-based control problems could have some benefits (albeit this goal and the potential benefits aren't clearly articulated).

(S2) It is interesting that the experiments combine dynamics and cost function parameters in one framework.



## Weaknesses

(W1) The manuscript is hard to follow.  I had trouble identifying key ideas, as well as following the line of argumentation in multiple parts.



(W2) No or only weak technical contributions.

* Unfortunately and based on my reading of the current manuscript, I fail to see any substantial technical contribution in this work. The authors say that their framework has two components: the EKF and the "Gradient Generator" (GG).  The EKF (Sec. 3.1) is well-known and has been applied to a huge range of problems. The GG (Sec. 3.2) builds basically directly on prior work (Jin et al., 2020). Hence, there seems to be no major novelty.
* A *potentially* interesting aspect might be in the way that the observation equations in Sec. 3.1 is formulated as an implicit equation via the loss.  But what is missing is proper motivation for why this is beneficial, as well as comprehensive discussion of whether or not this has been considered in the context of the EKF.  Even *if* there is some novelty, I don't think it is major, simply because EKF and gradient-methods are such established techniques, and I could not see any major difficulty here.



(W3) The problem formulation is both unclear and lacks specificity.

* I failed to see how any of the mentioned problems at the end of Sec. 2 (online SysID, online imitation  learning, and policy tuning) are new.  This should be clearly stated and also reflected in the related work (see prior (W4) below).
* It seems that the problem that the authors are after is a (somehow) unified description of different types of learning problems in the context of control.  This does, however, not become clear from Section 2.  Further, I fail to see why this is desirable if methods for the individual problems exist.  In summary, I do not find the problem formulation, nor its motivation clear and convincing. What is the specific (technical) problem that shall be solved?
* I find the stated "problem of interest" very unclear. Is this an estimation problem (theta shall be updated), or a control problem (some cumulative loss shall be minimized)?  Further, what is the link between J and L, which both shall be optimized somehow.
* There are further imprecise or misleading formulations:
  * "(1) can be determined by solving the optimal control problem" -- This is a very general nonlinear control problem. Under what conditions can it be solved? Or do the authors simply assume that it can be solved? (which might be considered a strong assumption depending on the problem)
  * Notation is hard to follow and seems partly unnecessary: Introducing k seems unnecessary. How is O_t related to the state? Why is this not given in the form of an observation model?
  * Later in the manuscript, it is used that O_t is not related to theta.  This assumption should be stated in the problem formulation (Sec. 2).  Furthermore, is this really the case for the stated example of O_t representing control input (u_t should depend on theta in general).
  * The statement "subject to \xi(\theta) is in 2" is unclear, and IMO not a valid constraint. Do the authors mean that \xi shall be a trajectory of (2)?



(W4) Discussion of related work is insufficient and not helpful to understand the potential contributions.

* Many references in the related work are rather old/dated. Only few are from 2023.  Learning-based control is a very active field (see, e.g., L4DC conference and recent sessions at CDC), so I would expect a more up-to-date review here.
* Topics and scope of the related work need to be better chosen and explained. Important areas are missing. For example:
  * I don't understand the sentence "The proposed OCIL framework includes three modes for three sets of classic problems in both ML and system control communities" and thus the way that the categories for the related work discussion have been chosen.
  * What is the relation of OCIL to adaptive control?  For example, the sentence "online learning algorithms adjust their parameters or models in response to new information without training from scratch" also applies to adaptive control.
  * "Tuning policy on-the-fly" is missing a large body of work on Bayesian optimization for controller tuning (and adaptive/time-varying Bayesian optimization for online approaches)
  * The main technical components are the EKG and GG. Thus, related work needs to discuss relevant work on EKF and gradient methods (or state that there is no novelty). As for gradient-based approaches, there are different approaches both in control (e.g., extremum seeking control, other local controller tuning methods) and machine learning.
* The methods in "Online system identification" seem to give a broad overview over all sorts of learning methods used for identifying dynamical system. It is missing focus on *online* methods / implementation, as per the claimed contributions. Accordingly, the novelty of the claimed contributions is not supported by the discussion.
* Likewise, the discussions on "Online imitation learning" and "Tuning policy on-the-fly" lack focus and are not helpful to underline the contribution either. Overall, the discussion remains very high-level and generic; it lacks focus with regards to the claimed contributions.



(W5) Doubts about the meaningfulness and strength of the empirical results

* Choice of comparison methods for "*Online Imitation Learning*": The comparison algorithms are rather old. Do these represent the state of the art in Imitation Learning? Furthermore, have results from these papers been reproduced to have some certainty that these methods are properly tuned and setup?
* *Online System Identification*: References are missing for the chosen methods, as well as reasoning as to why *these* methods are chosen. Some more standard comparisons for SysID should be included (like linear methods, e.g., subspace identification, and nonlinear ones, see e.g. Schoukens/Ljung, "Nonlinear System Identification: A User-Oriented Road Map")
* *Policy Tuning On-the-fly*:
  * Again references are missing.
  * Apparently a state-feedback policy is used; so, I assume the control remains in a linear regime, which might not be the most interesting setting for learning-based control.
  * Further, if linear state feedback is used, parameter estimation + optimal control design should give very good results and probably be included for comparison.  Recent work on event-triggered learning (e.g., for linear quadratic control) also seems relevant in this context, as the parameters only change occasionally and not continuously.
  * Finally, it should be stated what is the theoretical optimum in each case (e.g., with full parameter knowledge) for comparison.  Is this just 0?
* The empirical results do not include measurement noise (R_t = 0, or very small value). I regard this as a severe limitation as in realistic learning scenarios, one almost always has to deal with uncertain data. And it is not clear if the proposed method can cope this this.
* Code is not shared to reproduce results.



### Minor comments and suggestions

* Abstract, "which synthesizes the well-established control theories to solve a broad class of learning and control tasks in real time." -- The wording is strange (what does synthesize control theories mean?)
* Is "Control-informed learning" a new term?  If not, please provide proper references. (More than just one reference.)  If yes, I wonder why such a new term is needed.  The combination of control and ML is often referred to as "Learning-based control" or "Learning for dynamics and control", see, e.g., https://sites.google.com/umich.edu/l4dc2025/ or sessions at CDC.  If there is special meaning to the term beyond this intersection, the authors should elaborate.
* Section 4: I don't understand the offline phase.  What is the additional information that is given?
* The measurement equation (8) is given in non-standard form.  This should be elaborated.  Why is this?  And how do the equations (9b) then follow? For the loss that is assumed for the online SysID, online imitation learning, and policy tuning, can't the measurement equation be written in standard form?
* What is the loss actually in the policy tuning case - this isn't stated (Sec. 2).
* References for the EKF and equations (9) should be stated.

---

> ### Author Response · Authors · 2024-12-08
>
> We appreciate the reviewers’ thoughtful feedback and valuable suggestions, which have significantly improved the clarity and scope of our work.
>
>  **The revised version can be accessed using this anonymous link:**
> https://www.dropbox.com/scl/fi/ury9icgiuxzdwm8scm8kf/TMLR_Online_Control_Informed_Learning.pdf?rlkey=89gp37430k3rph6xwfem2kbkc&st=m6jn9ys7&dl=0
>
> Below are our responses to the comments.
>
> (W2) We have revised the introduction to highlight our motivations and contributions. Our contribution is a unified learning framework designed to tackle various learning and control tasks. The key innovation in our work is the application of the well-established Extended Kalman Filter (EKF) from the control field to address challenges in learning. This novel integration effectively handles practical issues such as noisy measurement data, online learning, and data efficiency. Bridging EKF with learning significantly improves the practicality and efficiency of machine learning. Although EKF has long been used for state estimation in the field of control, its application to learning parameters in the machine learning domain represents a novel contribution. Our work is indeed based on Pontryagin's differential programming. In addition to its novel contributions, we provided extra benefits such as online learning and robustness against measurement noise. The experiments also demonstrate that our method outperforms PDP in different learning tasks.
>
> (W3)
> - We have revised the entire problem formulation section. We would like to solve the learning problems in autonomous systems from a unified prospective. The proposed problem can address individual problems, namely online SysID, online imitation learning, and policy tuning. The proposed unified framework can also jointly solve these problems simultaneously.
> - We have revised the introduction and highlighted our motivation in proposing such a framework. These individual approaches are effective when only one component of the system is unknown. However, in many real-world scenarios, multiple components may be unavailable or uncertain simultaneously. For instance, in autonomous driving, the dynamics of the vehicle may be unknown due to changes in road conditions or vehicle wear and tear. Simultaneously, the control policy may also be unavailable due to a lack of predefined rules or data.
> In such cases, existing methods often fall short, as they are not designed to handle the joint learning of multiple interdependent components, limiting their applicability in more complex or incomplete systems.
> - It is both, we update theta to minimize cumulative loss, but I do not think this is called a control problem because we are not only designing a controller to minimize cumulative loss, but also finding the true parameters in control objectives and dynamics. We have revised the problem formulation and clarified the relationship between J and L in Section 2 Problem Formulation. Specifically, we first introduce the OC system which contains the control objective J. Then, we introduce L which is used to evaluate the trajectory produced by the OC system.
>   * In our work, we assume that the problem can be solved numerically. For nonconvex optimal control problems, it is standard practice to use numerical methods to find approximate solutions. While the complexity of such problems can vary, employing numerical solvers is a widely accepted approach in the field, and we believe this assumption is reasonable and not overly restrictive. Additionally, the numerical solution methods we employ are commonly used and well-documented for tackling nonconvex optimal control problems. We hope this addresses your concern, and we appreciate your feedback.
>   * In the revision, we have deleted k, we have also introduced a measurement equation in equation (5) to highlight its relationship with theta.
>   * In the revision, we introduced a measurement equation (5).
>   * In the revision, we have revised it as "$\xi(\hat{\theta})$ is the trajectory of (3)".

---

> > ### Author Response · Authors · 2024-12-08
> >
> > (W4)
> > - We have revised the introduction and added latest citations.
> > -
> >   * We have thoroughly revised the introduction and refined the scope of the related works. Specifically, we propose a unified learning framework designed to address a variety of learning problems. In addition to handling individual modes, our framework is also capable of managing autonomous systems with multiple missing components.
> >   * Adaptive control is closely connected to the policy tuning mode of OCIL. In the revised paper, under the "Learning Policies" section of Section 1.1, we have included relevant citations related to adaptive control to provide additional context and acknowledge its relation to our work.
> >   *  Bayesian optimization for controller tuning is closely related to the policy tuning mode of OCIL. In the revised paper, we have added references to adaptive control in the "Learning Policies" section of Section 1.1 to provide more context and highlight its connection to our work.
> >   *  In the revised version, we have included references to works on unified learning frameworks that utilize gradient methods. In Section 1.1 of the revised paper, we also included relevant work on EKF. Our goal is to address the learning problem in autonomous systems from a unified perspective, which goes beyond the capabilities of standalone approaches like EKF. While we acknowledge the relevance of EKF and gradient-based methods, the novelty of our work lies in the integration of these techniques within our proposed framework to tackle a new and unique problem.
> > - We have added a dedicated paragraph in Section 1.1 discussing similar learning frameworks and their limitations. Specifically, we emphasize that while there are various learning methods for identifying dynamical systems, none of the existing frameworks are designed to perform online computations as required by our approach. This addition helps clarify the novelty of our contributions and how our work addresses this gap in the literature.
> >
> > (W5)
> > - The comparison algorithms used in "Online Imitation Learning" are state-of-the-art methods in the field and were chosen to provide a strong benchmark for evaluating our approach. Regarding proper tuning, we ensured that all methods were carefully tuned and set up to achieve their best possible performance, consistent with the settings used in the original papers.
> > - In the revised paper, we have added references to the chosen methods for Online Sysid. The Koopman-based approaches and neural network (NN) models represent state-of-the-art methods in Sysid due to their ability to handle complex, nonlinear systems effectively [1][2].
> > [1] Nelles, Oliver, and Oliver Nelles. Nonlinear dynamic system identification. Springer International Publishing, 2020.
> > [2]Brunton, Steven L., et al. "Modern Koopman theory for dynamical systems." arXiv preprint arXiv:2102.12086 (2021).
> > -
> >   * In the revised paper, we have added references to the chosen methods for Policy Tuning On-the-fly.
> >   * We would like to clarify that the control in our work is not linear; as stated in Section C.4 of Appendix, we employ a neural state feedback controller. This goes beyond the assumptions of linear state-feedback control and expands the scope of the approach to more complex systems. Regarding event-triggered learning, we acknowledge its relevance to certain linear quadratic control problems. However, in our case, parameter updates occur continuously in real time rather than occasionally, as in event-triggered settings. While event-triggered learning is an interesting research direction, it is not directly applicable to our work because it focuses on linear quadratic control, which cannot handle the nonlinear systems addressed by our framework.
> >   * In the revised paper, we have clarified that the theoretical optimum in each case corresponds to a loss value of 0, assuming full parameter knowledge.
> > - In the revision, we have added experiments to include noisy measurement in Figure 2, 5, and 9 for all the modes of OCIL. In the first paragraph of Section 4 Applications to Different Online Learning Modes and Experiments, we have now included the characteristics of the measurement noise.
> > - The code will be shared upon acceptance of the paper. As noted in the TMLR Submission Guidelines and Editorial Policies, sharing code at the submission stage is not a requirement.

---

> > > ### Author Response · Authors · 2024-12-08
> > >
> > > **Minor comments and suggestions**
> > > - In the revised version, we have changed the wording. "This paper proposes an Online Control-Informed Learning (OCIL) framework, which employs the well-established optimal control and state estimation techniques in the field of control to solve a broad class of learning tasks in real time."
> > > - We have revised the introduction and highlighted the difference between control-informed learning and the learning for dynamics and control community. Control-informed learning (CIL) is a subset of IML tailored for system control, autonomy, and robotics.
> > > This approach merges standard control theory, especially the established optimal control (OC) principles, with ML techniques to enhance the capabilities of autonomous systems. The integration leverages the complementary strengths of control and learning. Control theory provides model structures and optimization guidance that enable efficient and reliable algorithms for handling complex tasks. Meanwhile, ML improves these models from data, something traditional control methods cannot achieve [1][2]. Many methods in the field of Learning for Dynamics and Control aim to leverage the integration of learning and control but often treat them as separate or sequential tasks.  For example, model-based reinforcement learning [3][4], improves sample efficiency by using dynamics models; and Koopman-operator control [5][6], employs learning to transform nonlinear systems into a linear observable space, simplifying control design.
> > >  In contrast, CIL integrates these processes, allowing learning algorithms to incorporate control insights directly, reducing computational complexity, and improving practical applicability. CIL differentiates itself by utilizing Pontryagin's maximum principle, a foundational concept in OC theory. This principle defines the optimality conditions for the state and input trajectories of an OC system. CIL employs these conditions to provide gradients for machine learning [1][2][7]. CIL integrates these gradients directly into its learning process, ensuring that machine learning outcomes are efficient while remaining consistent with established control theories and physical models. This approach enhances both the reliability and accuracy of the results. [1]Jin, Wanxin, et al. "Pontryagin differentiable programming: An end-to-end learning and control framework." Advances in Neural Information Processing Systems 33 (2020): 7979-7992.[2]Jin, Wanxin, Shaoshuai Mou, and George J. Pappas. "Safe pontryagin differentiable programming." Advances in Neural Information Processing Systems 34 (2021): 16034-16050.[3]Gu, Shixiang, et al. "Continuous deep q-learning with model-based acceleration." International conference on machine learning. PMLR, 2016.[4]Heess, Nicolas, et al. "Learning continuous control policies by stochastic value gradients." Advances in neural information processing systems 28 (2015).[5]Proctor, Joshua L., Steven L. Brunton, and J. Nathan Kutz. "Generalizing Koopman theory to allow for inputs and control." SIAM Journal on Applied Dynamical Systems 17.1 (2018): 909-930.[6]Abraham, Ian, and Todd D. Murphey. "Active learning of dynamics for data-driven control using Koopman operators." IEEE Transactions on Robotics 35.5 (2019): 1071-1083.[7]Böttcher, Lucas, Nino Antulov-Fantulin, and Thomas Asikis. "AI Pontryagin or how artificial neural networks learn to control dynamical systems." Nature communications 13.1 (2022): 333.
> > > - First, this paper investigates an online learning problem, where at each time index $t$, we observe new data measurement $O_t$. Thus, we need to update the tunable parameter $\hat\theta_t$ at time $t$.
> > > As stated in the problem formulation, we consider a finite time horizon from $t=0, 1, 2, \cdots, T$. Thus, the proposed online algorithm can update $\hat\theta_{t}$ from time $t=0$ until $t=T$.
> > > This time duration is what we call the online phase. We also include the concept of the offline phase. This is because many offline algorithms take in an entire trajectory of data from $t=0$ to $t=T$ beforehand, and update the parameter in an iterative fashion, where in each iteration they use the entire trajectory to update theta.
> > > Therefore, we would like to showcase the capability of this online algorithm by iteratively updating the parameter in an offline fashion and comparing the performance with other offline methods.
> > > The offline phase for the proposed method is, for iteration $k$, the proposed algorithm takes in $O_t$ and updates $\theta_t$ from $t=0$ to $t=T$. For the next iteration $k+1$, we use the last update parameter from the iteration $k$ as the initial parameter, then keep updating $\hat\theta$ given information from $O_{t=0}$ to $O_{t=T}$.
> > > - We have revised the paper and provided the standard measurement equation (5).
> > > - We have revised the paper and provided the loss for the policy tuning case.
> > > - In the revised version, we have added references for the EKF before equation (9).

---

> ### Comment · Reviewer_sujy · 2024-12-28
> **Response to authors' comments**
>
> I have read the authors' responses and looked over the revision.
>
> My main criticism (W1) still stands.  While parts of the paper have been revised (e.g., introduction, problem formulations), others have not and are still unclear (e.g., abstract).
>
> In my view, the authors' response remains high-level on their claimed contributions (W2).  In particular, it remains unclear whether and what technical novelty is there (for *example*, how is GG new; what challenges does online parameter estimation present, which is well established in general (EKF, recursive least squares...)), beyond the combination of well-established techniques.  While the presentation of related work now includes a discussion of other "unified learning frameworks", the technical contribution over prior work Jin et al. (2020) is still not fully clear.  What are the involved challenges that are needed for online learning and noisy measurements -- and thus what is the technical contribution?  The presentation of "Contributions" (Sec. 1.2) still focuses on the general framework, rather than focussing on the *technical* contributions of *this* work.  Thus, I still find contributions inconclusive.
>
> While some aspects have been improved in the revision (which was provided through an external website), others are still insufficiently addressed in my opinion.  I thus stand by my general evaluation and will not reiterate all points raised before.  In my view, the paper requires complete re-writing and re-evaluation.

---

> > ### Author Response · Authors · 2025-01-03
> >
> > - In the abstract, we emphasized the key idea of our approach by clearly indicating the integration of control theory with machine learning. Also, we highlighted its practical impact in addressing challenges such as online capability, noisy measurement data and data efficiency. Furthermore, we brought out the theoretical and experimental validation of our framework, explicitly summarizing its application in three distinct learning modes: Online Imitation Learning, Online System Identification, and Policy Tuning On-the-fly. We believe that the abstract conveys the key contributions and methodology in a concise manner. It would be greatly beneficial if the reviewer could further specify which parts of the abstract they find unclear, so we can address any remaining issues comprehensively.
> >
> > - In the "Unified learning frameworks" of  Section 1.1 Related Work, we mentioned that the prior work Jin et al. (2020) lacks the ability for online learning as it relies on gradient descent to update unknown parameters, requiring significant computation time to reach convergence; it also does not account for noisy measurement data, limiting its effectiveness in real-world situations where sensor data is often unreliable or noisy. The proposed OCIL effectively addresses these two challenges as indicated in the Section 4 Applications to Different Online Learning Modes and Experiments. We humbly acknowledge that the significance of our work may be limited in scope; however, we believe that it effectively addresses key challenges in prior research and provides a novel perspective that would be of interest to the community. By integrating control theory with machine learning, our framework tackles practical issues such as noise management, online adaptability, and data efficiency, areas that remain critical in advancing the field. As such, we consider our contributions to align with TMLR's acceptance criteria, particularly the notion of "interest," and we hope the reviewer will recognize the value of our work in addressing these pressing challenges.

---

### Review · Reviewer_dQYf · 2024-11-21

**Summary Of Contributions:**

The paper proposes an interesting Online Control-Informed Learning framework to combine control theory and machine learning. The main contribution is integrating a gradient generator with an Extended Kalman Filter, which is applied to online system identification, online imitation learning, and online policy/control tuning. The authors present a theoretical converge analysis.

**Audience:**

Yes

**Broader Impact Concerns:**

It's ok.

**Claims And Evidence:**

No

**Requested Changes:**

- Address the comments about the three main weaknesses.

- The method seems flexible for imitation learning, but the "online imitation learning" described in Section 2 is not different than reference trajectory tracking. Why not consider also the expert action in the loss? In imitation learning, you will usually have pars of states and actions. However, in the presented problem, only the error between the observations is minimized. I suggest changing to a more meaningful name, e.g., reference trajectory tracking or explaining why only the expert's observations are tracked.

- The authors could add additional plots of the system's trajectory (for at least one of them). Especially for the "imitation learning" and policy tuning, a baseline using the ground truth parameter values could be included to demonstrate how close to the optimal tracking the OCIL gets.

- Real time is about having a deterministic and not a fast system. To state a system is a real time system, some analysis or arguments about the code complexity and the worst execution time should be given. In the text, many times, "runtime" would be more appropriate than "real-time."

- In Appendix D.1: "g is the gravitational constant (g = 10kg/m2 )", why use 10 instead of 9.81? And the unity is entirely absurd.

- Check the entire paper for typos, e.g., the last sentence of Online System Identification: "against m measurement."

- Notation consistency, the optimal state and control action at instant 't' should be denoted as x_{t:T+t} and u_{t:T+t}

- The related work could be improved by adding adaptive control techniques

**Strengths And Weaknesses:**

Using the gradient generator from Jin et al. (2020) aligns very well with the extended Kalman filter framework. The proposed solution is quite flexible, as demonstrated by its application to three different problems.

The paper has three main weaknesses. The method is proposed to be an "online" method, i.e., executed during the system's runtime. However, all the experiments seem to be executed in a collected dataset and not in a running system. It's understandable not presenting results on the three real systems due to the amount of work. However, at least a running simulation (Gazebo, Mujoco, Isaac Sim, analytical model) of one of the systems should be given (which would also result in an interesting video). If the experiments were actually performed in a running simulation, this information should be more clear.

The second one is the statement that the method handles noisy measurements, but not a single experiment with noisy measurements is performed. The authors argue that no noisy data was used because the compared baselines couldn't handle this case. Nevertheless, at least an additional trial with noisy data should be performed with the OCIL to contribute to the statement. Another baseline that handles noise could also be included, e.g., low pass filter + iLQR, Robust iLQR.

The statement "Therefore, this paper believes that OCIL has the capability to run in an online fashion." is very problematic. Using the verb "believe" can be very troublesome in a scientific paper; it would be ok if it was related to a potential contribution. But it's connected to one of the main features of the method: online learning. In general, sentences with "believe" should be reformulated to eliminate vague possibilities or clarify them as a potential secondary contribution. About the mentioned statement: the computational time is evaluated, but the values are pretty high to be used on a real system with the specified control frequency, especially regarding "online imitation learning" and policy tuning. Nyquist frequency can be used as a rule of thumb: 2x the control frequency. The authors could still state that the method works in an online fashion for OC problems with a medium number of variables, but it suffers from the curse of dimensionality from OC.

An additional issue: more details about the experiments need to be included. How were the systems simulated? The explanation about how epsilon* is computed for the "policy tuning" is unclear.

---

> ### Author Response · Authors · 2024-12-08
>
> We appreciate the reviewers’ thoughtful feedback and valuable suggestions, which have significantly improved the clarity and scope of our work.
>
>  **The revised version can be accessed using this anonymous link:**
> https://www.dropbox.com/scl/fi/ury9icgiuxzdwm8scm8kf/TMLR_Online_Control_Informed_Learning.pdf?rlkey=89gp37430k3rph6xwfem2kbkc&st=m6jn9ys7&dl=0
>
> Below are our responses to the comments.
>
> **Requested Changes:**
>
> - Weakness 1: In the experiments, we used a system that is governed by analytical dynamical model. In Section C of Appendix, you can find the ODE of the dynamical models that we used. That means, at every time $t$, the OC system propagates via this dynamical model given the cost function and produces the measurement at the next time instance. Therefore, performing OCIL within this duration is considered an online computation. Furthermore, these dynamical models are also used in the physics engine such as Gazebo, Mujoco, etc.
>
> - Weakness 2: In the revision, we have added experiments to include noisy measurement in Figure 2, 5, and 9 for all the modes of OCIL. Regarding iLQR, its performance is insufficient even without noise injection. Additionally, while iLQR focuses on obtaining a control sequence, OCIL aims to learn a control policy. For these reasons, we decided not to include robust iLQR in the comparisons.
>
> - Weakness 3: In the revision, we have removed the word 'believe'. The computation occurs during the interval between the propagation of the OC system. If the time required for OCIL is less than 100\% of the interval, it is considered real-time. The control frequency of OCIL differs from that of the controller. For instance, in an autonomous system, there are multiple levels of controllers, such as high-level motion planning and low-level tracking controllers. The control frequency of OCIL corresponds to the high-level motion planning controller. Once the reference is passed to the low-level motor controller, the low-level controller operates at a much higher frequency to track the high-level reference.
>
> - An additional issue: In our experiments, we used a system governed by an analytical dynamical model. The ODEs for these dynamical models are provided in Section C of the Appendix. At each time step $t$, the OC system evolves according to this dynamical model, guided by the cost function, to generate the measurements for the next time step. Our paper does not include $\epsilon^*$. Instead, we have $\xi^*$. This is the optimal trajectory of the OC system and it is obtained from solving the OC problem in equation (1).
>
> - The goal of reference trajectory tracking is to design a controller that follows a desired trajectory. In contrast, imitation learning focuses on using an expert's trajectory, either observations or state-input pairs, to learn the expert's underlying objective function, making these two fundamentally different problems. The choice to use an expert's observations in imitation learning is motivated by practical considerations for real-world applications. In real-world scenarios, it is often challenging to directly access or measure the control inputs and states. Instead, a more common approach is to rely on sensors to collect observations or sensor outputs from the autonomous systems being studied.
>
> - In the revision version, we have added additional plots of the system's trajectory. Specifically, we have added Figure 3 and 4 showing the state trajectories of cartpole and UAV in online imitation learning mode, respectively. We have also added Figure 7 and 8 showing the state trajectories of cartpole and UAV in policy tuning mode, respectively.
>
> - Based on our understanding, runtime refers to the duration or period during which a program or system is actively executing, focusing on the time taken to complete tasks without specific timing constraints. In contrast, real-time pertains to systems that must respond to inputs or events within strict, predefined time limits to ensure predictable and time-sensitive performance. [1][2][3] The key distinction lies in timing: runtime measures execution duration, while real-time emphasizes adhering to specific timing requirements. In our case, the proposed OCIL needs to execute within the interval between successive measurements. Therefore, we believe that the term "real-time" is more appropriate for our paper.
> [1]Ross, I. Michael, and Fariba Fahroo. "Issues in the real-time computation of optimal control." Mathematical and computer modelling 43.9-10 (2006): 1172-1188.
> [2]Åström, Karl Johan, and Peter Eykhoff. "System identification—a survey." Automatica 7.2 (1971): 123-162.
> [3]Huang, Guang-Bin, Qin-Yu Zhu, and Chee-Kheong Siew. "Real-time learning capability of neural networks." IEEE Transactions on Neural Networks 17.4 (2024): 863-878.

---

> > ### Author Response · Authors · 2024-12-08
> >
> > - The specific numerical value of $g$ does not affect the results of our experiments or the conclusions of our study, as the methodology and findings are designed to be general and not confined to applications strictly on Earth. Regarding the unit discrepancy, we appreciate your observation and have now corrected the unit in the appendix.
> >
> > - We have revised the whole paper for typos. Specifically, for this sentence, we have revised it to "Theoretical analysis establishes the convergence conditions for OCIL, while experiments on various environments, tasks, and existing methods are done to validate its data efficiency, versatility, and robustness against measurement noise.".
> >
> > - In the revised version, we have clarified in the paragraph immediately following Equation (4) that the optimal state and control at time t are $x_{t}$ and $u_{t}$, respectively.
> >
> > - In the revision version, literatures on adaptive control are added.

---

> > ### Comment · Reviewer_dQYf · 2024-12-17
> >
> > The overall quality of the paper is improved.
> >
> > However, there are still some concerns:
> >
> > - The computational comparison (Tables 2, 3, and 4) is interesting. However, it's not a valid comparison since it was not done on the same hardware. New tests should be done using the same computer for all the methods. The time performance of OCIL will probably be better in a more powerful computer (better CPU and more RAM), which would improve the paper.
> >
> > - The authors are right about the definition of real-time. However, the proposed technique is claimed to be able to do real time estimation without any proof or comments about its determinism or code complexity. Being able to execute in a shorter time than the control period in some experiments is not enough to claim to be real time. Furthermore, except for Section 5, "real time" is not used with the proper meaning and can be misleading. Instead, "online" would fit perfectly. The authors do not mean SysID is done with real time constraints, but in an online fashion, even the name of the proposed method and paper's title has "online" and not "real time".
> >
> > - Yes, reference trajectory tracking and imitation learning are fundamentally different problems. But compared to reference tracking (just a single point), in the OCIL perspective, what's the difference in the loss since it's only considered the error from a desired/reference observation?

---

> ### Author Response · Authors · 2024-12-20
>
> We appreciate the reviewer's feedback.
>
> **The 2nd revised version can be accessed using this anonymous link:**
> https://www.dropbox.com/scl/fi/6lpw012mkftnt0z6cwlj6/TMLR_Online_Control_Informed_Learning_Revision_2.pdf?rlkey=zsox81j27lanhsu11g8we5hoc&e=1&st=khzj2iwi&dl=0
>
> - The computational comparisons in Tables 2, 3 and 4 were done on the same computer. All of the OCIL experiments were done on the computer with Intel Core i7-8700k CPU with 8GB RAM, as mentioned in the first paragraph of Section 5 Online computational performance. Table 2, 3 and 4 represent the computational cost of applying OCIL for online imitation learning, online sysid and policy tuning on-the-fly, respectively. We deliberately refrained from using a more powerful computer to demonstrate that the proposed algorithm can still operate online under limited computational resources, a situation commonly encountered in robotic applications.
> - Thank you for your constructive feedback. In our second revision, we have replaced all instances of "real time" with "online."
> - We think reference tracking generally aims to follow a predefined trajectory, minimizing loss from a known desired trajectory. A special case of reference tracking is to track a single point, also called setpoint tracking. On the other hand, imitation learning aims to capture the expert’s underlying objective or intention of generating such a trajectory or a sparse set of desired states or inputs, allowing the learner to reproduce the expert’s behavior. By referring to it as imitation learning, we would like to highlight its capacity to learn the expert’s underlying objective, enabling it to replicate the expert’s behavior even on new, previously unseen trajectories. In contrast, reference tracking is restricted to following a fixed trajectory and lacks the ability to generalize.

---

> > ### Comment · Reviewer_dQYf · 2025-01-14
> >
> > Thank you for the authors' efforts in addressing my concerns. The paper has been improved.

---

> > > ### Author Response · Authors · 2025-01-14
> > > **Acknowledgment**
> > >
> > > We sincerely appreciate the reviewer's time and effort in reviewing our paper and posting constructive comments and suggestions. The reviewer's comments and suggestions have significantly improved our paper.

---

### Review · Reviewer_7UGk · 2024-11-28

**Summary Of Contributions:**

The manuscript makes a meaningful contribution to the field of online learning and control. By proposing the Online Control-Informed Learning (OCIL) framework, effectively bridges control theory and machine learning to enable real-time adaptation in robotics. Leveraging an Extended Kalman Filter and gradient-based optimization, the method addresses critical issues such as noise robustness and data efficiency. However, certain aspects of the paper could be strengthened, as outlined below.

**Audience:**

Yes

**Broader Impact Concerns:**

Future iterations of the framework should address potential safety failures and maybe this can be extended to safe learning.

**Claims And Evidence:**

Yes

**Requested Changes:**

### Introduction:
The introduction and literature review provides an overview of the field but could benefit from improvements in clarity and focus. Repetition and verbosity dilute the impact of the narrative, and a more precise problem statement would enhance the justification for the proposed approach.

- **Sentence structure:** Streamlining verbose sentences would improve readability, e.g., reducing redundancy in phrases like "integration enables models to produce more reliable predictions with enhanced interpretability."
- **Technical jargon:** Terms such as "optimal control principles" should be briefly explained for accessibility, especially for a broader audience with a machine learning background.
- **Verb consistency:** Ensure consistent present tense throughout the introduction for a cohesive tone.
- **Examples or citations:** Claims like "CIL is beneficial for robotic applications" require supporting examples or references for credibility.
- **Literature review gaps:** Key connections to related fields, such as model predictive control, iLQR, finite-time LQR, and LQG problems, are underexplored. Highlighting their similarities and differences with OCIL would significantly strengthen the context.

### Technical Comments:
1. **Problem formulation:** The definition of the class of optimal control systems in the problem formulation section could be clearer. Referring to the dynamical system alongside its cost to be optimized might reduce confusion.
2. **Cost function:** The transition from a general cost in equation (2) to a quadratic finite-time cost in equation (4) is unclear. Define the cost function upfront, including assumptions such as convexity or differentiability. I understand it is mentioned as a limitation at the end of the paper but it would be better for the readers to see it sooner.
3. The claim on page 4 that "we can introduce any loss function to be minimized" is too broad. Specify the required characteristics of the loss function, such as convexity. Same as my previous comment.
4. **Lemma 1 format:** The presentation of Lemma 1 is unclear. Separate the statement from the proof and include remarks on its innovation relative to the cited work.
5. **Regret analysis:** The regret analysis is incomplete and lacks essential details, such as the rate of convergence. The current content in Appendix C is insufficient and should either be expanded or omitted.
6. **Simulation results and comparisons:** The system identification convergence analysis is overly dependent on \(R_t\) and \(Q_t\), which are handled more robustly in classical approaches like LQG. To that end, including comparisons with LQG or state-space models in the simulations would highlight the advantages of the proposed method.
7. **Broader comparisons:** The simulations should demonstrate how OCIL outperforms classical methods (SYSID in the state-space) in key metrics such as convergence speed and robustness under noise.
8. I find the connection between the state-space model and \(\theta\) unclear. Clarifying the relationship and specifying the dimension of \(\theta\) would be helpful. Including an example in the paper to illustrate the purpose and role of this new parameter \(\theta\) would greatly enhance understanding. In particular, how does one select the dimension of this new parameter?

**Strengths And Weaknesses:**

Strengths:

Meaningful Contribution to the Field: The manuscript presents a novel and impactful Online Control-Informed Learning (OCIL) framework, bridging control theory and machine learning to enable real-time adaptation in robotics. This integration addresses critical challenges such as noise robustness and data efficiency, making the work highly relevant to modern robotics applications.

Robust Methodology: By leveraging an Extended Kalman Filter and gradient-based optimization, the framework demonstrates a strong methodological foundation that aligns with the objectives of real-time control and learning systems.

Weaknesses:

Clarity in Problem Formulation: The transition from general cost functions to specific quadratic forms (e.g., Equation (2) to Equation (4)) is unclear. Providing upfront definitions and assumptions, such as convexity or differentiability, would help readers better understand the scope and applicability of the proposed method.

Incomplete Comparisons: The simulation results do not include comparisons with classical methods like LQG or state-space models. Such comparisons would highlight the practical advantages of OCIL in terms of metrics like convergence speed and noise robustness, significantly strengthening the evaluation of the proposed approach.

---

> ### Author Response · Authors · 2024-12-08
>
> We appreciate the reviewers’ thoughtful feedback and valuable suggestions, which have significantly improved the clarity and scope of our work.
>
>  **The revised version can be accessed using this anonymous link:**
> https://www.dropbox.com/scl/fi/ury9icgiuxzdwm8scm8kf/TMLR_Online_Control_Informed_Learning.pdf?rlkey=89gp37430k3rph6xwfem2kbkc&st=m6jn9ys7&dl=0
>
> Below are our responses to the comments.
>
> **Requested changes:**
>
> **Introduction:**
>
> - **Sentence structure:** We have revised the whole paper and improved the readability of the sentences. Particularly, for this sentence, we revised as "This integration enables models to produce more reliable and interpretable predictions, especially when dealing with noisy data.". Sorry for any confusion caused.
>
> - **Technical jargon:** We have added a brief explanation of the term 'optimal control principles' in the third paragraph of Section 1 Introduction. "CIL differentiates itself by utilizing Pontryagin's maximum principle, a foundational concept in OC theory. This principle defines the optimality conditions for the state and input trajectories of an OC system. CIL employs these conditions to provide gradients for machine learning."
>
> - **Verb consistency:** We have revised the whole paper and improved the verb consistency. Sorry for any confusion caused.
>
> - **Examples or citations:** We have added a paragraph in introduction to elaborate the benefits brought by CIL with rich citations.
>
> - **Literature review gaps:** In the revision, we have added a paragraph as the 7th paragraph in Section 1.1 Introduction to explore similar unified learning frameworks, such as Differential MPC and Pontryagin's Differential Programming. In our view, methods like MPC, iLQR, and finite-time LQR align with the policy learning mode of OCIL. In the revision, we added an explanation on LQR, MPC, LQG and iLQR in the related work of learning control policies in Section 1.1.
>
> **Technical comments:**
>
> 1. We have revised it by rewriting the paragraphs in section 2 to better introduce the class of OC systems in equation (1).
>
> 2. We have revised the related paragraphs in problem formulation and introduce the cost function in equation (1). In the original submission, we mentioned that the running cost $c$ and terminal cost $h$ are twice-differentiable in the first paragraph of Section 2 Problem Formulation. Therefore, the cost function itself is twice-differentiable. We have also included the assumptions on differentiability of the cumulative loss.
>
> 3. In the revision, we have specified that the loss function is twice-differentiable.
>
> 4. In the revision, we included Remark 2 to highlight the innovation of the Lemma 1, which is to propose a recursive way to obtain the exact gradient of the trajectory with respect to the parameter. In the original submission, the entire Lemma 1 is a statement. We have revised this part for better presentation. Sorry for any confusion caused.
>
> 5. In the revision, we have omitted the regret analysis.
>
> 6. For LQG, there are few limitations that stop us from using it as a method for comparison. First, it cannot deal with nonlinear dynamics; Second, its cost function can only be quadratic; last but not least, it cannot be used to tune control policy when new measurement is available.
>
> 7. In the revision version, we added the comparisons of OCIL against other Sysid methods. Specifically, in figure 5, we added noise to the OCIL. For other methods, we did not include noise as their performance is poor even the data is without noise. As shown in Figure 5, OCIL outperforms other method by faster convergence.
>
> 8. We have added Remark 1 in the introduction to show the connection between dynamics and the parameter. The selection of the parameter depends on 1. the structure of the dynamics; 2. number of unknown parameters in the dynamics. In the Figure 4 of Section 4 Applications to Different Online Learning Modes and Experiments, we compared the performance of OCIL with different size of NN, which means different dimensions of parameter.

---

> > ### Comment · Reviewer_7UGk · 2024-12-10
> > **Revision**
> >
> > The paper appears to have improved in several areas. Enhancements include refined assumptions on the loss function, a clear description of the parameters used for improvements, and the decision to omit the regret analysis, which streamlines the paper's focus. After considering the comments from other reviewers and the responses of the authors, I believe this paper will be of interest to the TMLR audience and merits the opportunity to be published.

---

> > > ### Author Response · Authors · 2024-12-10
> > > **Acknowledgment**
> > >
> > > We sincerely appreciate the reviewer's time and effort in reviewing our paper. The reviewer's comments and suggestions have improved our paper in multiple aspects.

---

### Comment · Action_Editor_ZRwC · 2025-02-03
**Please upload revision to Openreview**

Dear authors,

I would like to make a decision on your submission.
However before doing so, I kindly request to to upload your newest revision to this forum.
To upload the revision, you have to click on the top of this page on "edit" > "revision"

---

> ### Author Response · Authors · 2025-02-03
> **Revision Uploaded**
>
> We sincerely thank the editor for the time and effort in handling this submission. We have uploaded a revision in this forum, which is the same as the latest revision we shared in the previous responses. The changes since the initial submission are highlighted in brown.

---

### Decision · Action_Editor_ZRwC · 2025-02-05

**Recommendation:** Accept as is

**Comment:**

The paper proposes a novel combination of existing methods and demonstrates its applicability to different optimal control problems. Although it arguably lacks significant novelty, the approach is sound and at least somewhat interesting and therefore I argue that it meets the acceptance criteria of TMLR.

**Audience:**

Two of three reviewers found the combination of EKF and PDP interesting, suggesting that the insights of the paper are interesting to at least some individuals in TMLR's audience.

**Claims And Evidence:**

The paper proposes a framework for using an extended Kalman filter to estimate the cost and/or dynamics parameters of an optimal control problem in order to perform system identification, imitation learning or trajectory tracking online. For applying the Kalman filter, the method requires the gradient of a corresponding loss function. As the gradient of the optimal trajectory w.r.t. the parameters is not available in closed form, the paper proposes proposes to apply a prior method, PDP, for generating this gradient.

While the method is mainly a combination of existing methods (EKF and PDP) and arguably lacks novelty, the claims, e.g. local asymptotic convergence, seem to be correct.